# The Kaposi’s Sarcoma-Associated Herpesvirus Protein ORF42 Is Required for Efficient Virion Production and Expression of Viral Proteins

**DOI:** 10.3390/v11080711

**Published:** 2019-08-02

**Authors:** Matthew Butnaru, Marta Maria Gaglia

**Affiliations:** 1Graduate Program in Biochemistry, Sackler School of Graduate Biomedical Sciences, Tufts University, 136 Harrison Ave, Boston, MA 02111, USA; 2Department of Molecular Biology and Microbiology, Tufts University School of Medicine, 136 Harrison Ave, Boston, MA 02111, USA

**Keywords:** KSHV, ORF42, late gene expression, herpesvirus, herpes UL7 family

## Abstract

Kaposi’s sarcoma-associated herpesvirus (KSHV), the etiologic agent of Kaposi’s sarcoma and other aggressive AIDS-associated malignancies, encodes over 90 genes, most of which are expressed only during the lytic replication cycle. The role of many of the KSHV lytic proteins in the KSHV replication cycle remains unknown, and many proteins are annotated based on known functions of homologs in other herpesviruses. Here we investigate the role of the previously uncharacterized KSHV lytic protein ORF42, a presumed tegument protein. We find that ORF42 is dispensable for reactivation from latency but is required for efficient production of viral particles. Like its alpha- and beta-herpesviral homologs, ORF42 is a late protein that accumulates in the viral particles. However, unlike its homologs, ORF42 appears to be required for efficient expression of at least some viral proteins and may potentiate post-transcriptional stages of gene expression. These results demonstrate that ORF42 has an important role in KSHV replication and may contribute to shaping viral gene expression.

## 1. Introduction

Kaposi’s sarcoma-associated herpesvirus (KSHV) is a gamma-herpesvirus and the etiological agent of Kaposi’s sarcoma (KS), as well as two rare lymphoproliferative diseases (primary effusion lymphoma and the B cell variant of multicentric Castleman’s disease) and KS-associated herpesvirus inflammatory cytokine syndrome [1,2,3,4]. KS develops predominantly in long-term immunosuppressed individuals, such as AIDS and transplant patients. In addition, KS was endemic in sub-Saharan Africa prior to the AIDS epidemic, where it remains highly prevalent and one of the major causes of cancer deaths [5]. KSHV establishes a long-term latent infection in patients, with periodic reactivation of the lytic cycle that results in virus production. Both latent infection and lytic replication are important for KS tumorigenesis. For example, treatment of KS patients with ganciclovir, an inhibitor of herpesvirus lytic replication, causes tumor regression [6]. Although these findings suggest that targeting lytic replication could reduce tumor formation, many aspects of the lytic cycle of KSHV are poorly characterized, including the function of many viral proteins. This limits the ability to develop approaches to reduce replication.

Many of the herpesviral genes have apparent homologs in all subfamilies of herpesviruses (alpha, beta and gamma) and are considered “core” herpesviral genes. The proteins encoded by these genes are thought to have conserved and essential functions in herpesviral replication. However, in some cases, even herpesviral genes that are clear homologs based on sequence analysis have evolved additional functions in specific viruses or subfamilies. An example is the alkaline exonuclease, which also functions as a host shutoff ribonuclease in gamma-herpesviruses, but not other virus subfamilies [7]. Nonetheless, in KSHV many of these “core” genes are annotated solely based on their known function in other herpesviruses. One of these proteins is ORF42, which has homologs in all herpesvirus subfamilies based on phylogenetic analysis [8], although the sequence identity among the homologs is limited (less than 20% across subfamilies). The function of ORF42 in KSHV is unknown, but based on findings in alpha- and beta-herpesviruses [9,10,11], it is classified as a component of the tegument, a layer of proteins in the virion. Null mutations of the ORF42 homologs UL103 in the beta-herpesvirus human cytomegalovirus (HCMV) and UL7 in the alpha-herpesviruses pseudorabies virus (PRV) and herpes simplex virus 1 (HSV-1) result in a decrease in virus production [10,12,13,14,15,16]. This attenuation is thought to arise from defects in virion formation and potentially directly in viral egress [12,13], although the exact function of UL103 and UL7 in these processes remains unclear. Other aspects of the viral replication cycle, such as DNA replication and viral protein expression, appeared unaffected by UL7 and UL103 mutations [12,13]. Additionally, in alpha- and beta-herpesviruses UL7 and UL103 are consistently found in the tegument, leading to the classification of their homologs as tegument proteins, although the data for gamma-herpesviral virions are less clear [9,10,11,17,18,19,20,21,22]. Interestingly, there is evidence that UL7 and UL103 may have additional biological activities. One study reported that virion-associated HSV-1 UL7 may regulate early lytic gene transcription during de novo infection [16]. Similarly, proteomic analyses of HCMV UL103-associated proteins revealed interaction with several cellular antiviral proteins including the DNA sensor IFI16 [23]. KSHV ORF42, as well as its gamma-herpesvirus homologs BBRF2 in Epstein Barr Virus and ORF42 in murine herpesvirus 68 (MHV68) [8], remain completely uncharacterized.

Here we report that KSHV ORF42 is a protein expressed with late kinetics and localized to the cytoplasm. Moreover, ORF42 is required for wild-type levels of viral replication, like its homologs in other herpesviruses. Interestingly, we found that ORF42 mutations reduced the levels of several proteins, including late proteins involved in virion formation. This suggests that part of the ORF42 function in KSHV is to promote viral protein expression, a function that may be unique to this virus or viral subfamily. Therefore, we have identified a new role for a conserved herpesviral protein in the replication cycle of KSHV and as a potential new regulator of viral gene expression.

## 2. Materials and Methods

### 2.1. Cells

HEK293T cells were cultured at 37 °C, 5% CO_2_ in Dulbecco’s modified Eagle’s medium (DMEM, Gibco, Thermo Fisher, Waltham, MA, USA) supplemented with 10% fetal bovine serum (FBS, Hyclone, GE Healthcare, Chicago, IL, USA). iSLK.RTA cells (kind gift of Dr. Jung) containing wild-type KSHV BAC16, BAC16 ORF42^PTC^, BAC16 ORF42^REV^, or BAC16 ORF42-Flag were maintained in DMEM supplemented with 10% FBS and 400 μg/mL hygromycin (Enzo Lifesciences, New York, NY, USA). KSHV-infected iSLK.RTA cells that also express a transgene (ORF42^PTC^, +ORF42, +ORF42-Flag and +empty) were maintained in DMEM supplemented with 10% FBS, 400 μg/mL hygromycin and 100 μg/mL zeocin (Invivogen, San Diego, CA, USA). Live cells were imaged using a Nikon eclipse TE2000-U.

### 2.2. Plasmids

pCDNA4/TO-ORF42-Flag (used for subcloning ORF42), pCDNA4/TO-C-terminal-Flag, and pJP1_Zeo were kind gifts by Dr. Glaunsinger. pJP1_Zeo is a modified version of the pLJM-EGFP vector (a gift from David Sabatini, Addgene #19319) [24], in which the puromycin resistance gene was substituted with a zeocin resistance gene. To express ORF42 from a transgene, ORF42 was PCR amplified from pCDNA4/TO-ORF42-Flag with or without the Flag tag and inserted between the AgeI and EcoRI restriction enzyme sites of the lentiviral vector pJP1_Zeo. To generate pCDNA4/TO-ORF42(untagged), ORF42 amplified from pCDNA4/TO-ORF42-Flag was re-inserted in the PmeI-digested pCDNA4/TO-ORF42-Flag vector. pJP1-Flag was generated by excising the GFP gene from pJP1_Zeo and replacing it with a 3xFlag tag using the restriction sites NheI and EcoRI. To generate the pCMV-ORF26 construct expressing the entire native mRNA, the gene locus was amplified by PCR from BAC16 and inserted into the backbone of pd2eGFP-N1 (Clontech, Takara Holdings Inc., Kusatsu, Japan) digested with NheI and AflII (to remove GFP and the SV40 3’UTR). T4 DNA ligase-based cloning (New England Biolabs, Ipswich, MA, USA) or Gibson cloning (HiFi Assembly Mix, New England Biolabs) were used to generate the constructs. Primers used in cloning are listed in Table 1.

### 2.3. Bacterial Artificial Chromosome (BAC) Mutagenesis and Generation of KSHV-Infected Lines

KSHV BAC16 was kindly provided by Dr. Jung and is described in Brulois et al. [31]. Genetic manipulation of the BAC was performed in GS1783 *Escherichia coli*, which encodes the Red recombinase and the restriction enzyme I-Sce, using the protocol described by Tischer et al. [32]. Primers used for mutagenesis are listed in Table 1. A kanamycin-resistance (Kan) cassette flanked by I-Sce cleavage sites was amplified by PCR. The primers introduced sequences at each end of the Kan cassette that correspond to those at the desired insertion/mutation point. These sequences are also homologous to each other. For the ORF42 premature termination codon (ORF42^PTC^) BAC, a Serine-to-stop mutation was inserted at amino acid 25 of ORF42 in the sequences flanking the Kan cassette. This mutation was designed so that the overlapping ORF43 605 amino acid reading frame would not be altered, because it causes a synonymous change in the Valine 585 codon (from GTC to GTG). The next in-frame methionine in the ORF42 coding sequence is at amino acid 58, which means that any translated product starting at this point would be severely truncated. For the ORF42 revertant (ORF42^REV^) BAC the wild-type sequence was added to the primers, and recombination was performed on the ORF42^PTC^ BAC. To construct the ORF42-Flag BAC, a Flag tag was added in-frame to the 3’ end of ORF42 coding sequence. Because the transcription termination and polyadenylation signal of ORF41, as well as a portion of the ORF41 coding region, overlap the 3’ end of ORF42 [25,33], the 3’ end of the ORF42 coding sequence was duplicated after the Flag tag to preserve ORF41 expression. This duplication also provides the transcription termination signal for ORF42, which overlaps the end of the ORF42 coding region. The sequence is shown in Appendix A. The fragment to carry out the BAC recombination was generated by fusing the 3’ end of ORF42 and the Flag sequence to the 5’ end of the Kan cassette. In addition, the Flag sequence and the 5’ end of the overlapping portions of ORF42 and ORF41 were fused to the 3’ end of the Kan cassette. The PCR fragments were electroporated into BAC16-containing *E. coli* GS1783 cells containing BAC16. Red expression was induced by heat shocking the cells at 42 °C in order to promote recombination of the homologous sequences. Recombinant clones were selected on Lysogeny broth (LB) plates containing 16 μg/mL chloramphenicol (Sigma-Aldrich, Merck KGaA, Darmstadt, Germany) and 25 μg/mL kanamycin (Fisher Scientific) at 30 °C. Insertion of the Kan cassette was confirmed by PCR. To remove the Kan cassette, I-Sce expression was induced by treatment with 1% l-arabinose, followed by induction of Red by heat shocking the cells at 42 °C. Kanamycin-sensitive clones were isolated by replica-plating on kanamycin-chloramphenicol and chloramphenicol-only arabinose-containing plates. The integrity of the genomic DNA isolated from these clones was verified by Restriction Fragment Length Polymorphism. The mutations/insertions were verified by PCR amplification of the region of interest and Sanger sequencing. After isolation of the BACs using the Nucleobond BAC kit (Machaery Naegel, Düren, Germany), iSLK.RTA clonal cell lines latently infected with the wild-type (WT), ORF42^PTC^, ORF42^REV^, and ORF42-Flag BACs were generated. Fugene was used to transfect the BACs into iSLK.RTA cells. Two days after transfection, 800 μg/mL hygromycin was added to select transfected cells. This procedure gave rise to clonal transfected colonies, which were expanded and tested for virus production and induction of ORF57 (an RTA target) after doxycycline addition. For our experiments, two colonies of WT KSHV-infected cells and one colony of KSHV ORF42^PTC^, ORF42^REV^ and ORF42-Flag infected cells were used. The KSHV ORF42^PTC^-infected cells were also transduced with Zeocin-resistant rescue constructs as described below. In early experiments differences in reactivation rates (ORF57 expression at day 2 post doxycycline addition), virion production and viral DNA replication among several clonal WT KSHV-infected lines were observed. Therefore, two lines of KSHV WT-infected cells were used as comparison for most experiments, to ensure that the reported phenotypes of KSHV ORF42^PTC^ were real differences between the WT and ORF42^PTC^ lines rather than natural variability in the WT phenotype. Since we analyzed only one clone for the ORF42^PTC^, ORF42^REV^ and ORF42-Flag recombinant viruses, we were unable to determine variability in the phenotypes of the recombinant viruses.

### 2.4. Lentiviral Transduction and Transient Transfection

To generate viral particles for lentiviral transduction, pJP1-based lentiviral expression plasmids were transfected with the packaging plasmids psPAX2 and pMD2 (gifts from Didier Trono, Addgene #12260 and 12259) into HEK293T cells. Two days later, BAC16-containing iSLK.RTA cells were infected with filtered supernatant by centrifugation at 1000 rpm for 1 h in the presence of 8 μg/mL polybrene (Sigma-Aldrich). Transduced cells were selected using 250 μg/mL zeocin. To express proteins and RNAs with transient transfection, constructs were transfected into HEK293T cells using linear polyethylenimine (PEI, Fisher Scientific). 1.25 μg/mL of DNA and 3.75 μg/mL of PEI were used to transfect cells in 6-well plates.

### 2.5. Virion Measurements by Flow Cytometry and qPCR

The lytic cycle was induced in 1.02 × 10^6^ KSHV-infected iSLK.RTA cells by addition of doxycycline (1 μg/mL). Six days post-induction the supernatant was collected and filtered to remove cells and other debris. HEK293T target cells were plated in 12-well plates at a density of 1.2 × 10^5^ cells/well and infected with virus-containing supernatant by centrifugation at 1000 rpm for 1 h in the presence of 8 μg/mL polybrene (Sigma-Aldrich). Three days post infection, cells were trypsinized and GFP-positive cells were counted by flow cytometry on a BD FACSCalibur at the Tufts Laser Cytometry core facility. The three-day window is needed to ensure clear GFP expression in the infected cells. FlowJo (BD Biosciences, San Jose, CA, USA) was used to analyze the data. Titer was calculated as: −N × ln (1−Ninf/N)/V, where N is the number of target cells at the time of infection, V the volume of supernatant (in mL) used to infect the cells and Ninf/N the fraction of GFP-positive infected cells measured three days post infection. This calculation is based on the assumption that the proportion of GFP cells will remain the same, even as cells divide. To isolate DNA from extracellular virions, 100 μL of supernatant were treated with 10 units of Turbo DNase (Ambion, Thermo Fisher) at 37 °C for 1 h to degrade extravirion DNA. The reaction was stopped by addition of EDTA (10 mM final concentration) and heating at 70 °C for 15 min. 0.4 mg of proteinase K and 200 µL of AL lysis buffer (DNeasy kit, Qiagen, Hilden, Germany) were added to lyse virions. The DNeasy kit was then used to isolate viral DNA following the manufacturer’s protocol. DNA levels were measured by qPCR using iTaq Universal SYBR reagent (Bio-rad, Hercules, CA, USA), using primers against the promoter for the latency-associated nuclear antigen (LANA) gene (Table 1) [25]. Levels were quantified against a standard curve generated by PCR amplification of LANA, in order to calculate the number of genomes per mL.

### 2.6. Measurement of Cell-Associated Virions

Cell-associated (intracellular) virions were isolated from infected cells using a protocol adapted from Sanchez et al. [34]. 1.7 × 10^5^ KSHV-infected iSLK.RTA cells were treated with 1 μg/mL doxycycline to induce the lytic cycle. Six days post-induction of the lytic cycle, the supernatant was removed, and cells were washed with phosphate-buffered saline (PBS) three times to remove extracellular virions. Cells underwent three freeze-thaw cycles in 2.5 mL DMEM supplemented with 10% FBS. The suspension of lysed cells was then filtered using a 0.45 μm filter to remove debris. HEK293T target cells were plated in 24-well plates at a density of 6 × 10^4^ cells/well. 2 mL of virus-containing suspension was added to each well in the presence of 8 μg/mL polybrene. Plates were then centrifuged at 1000 rpm for 1 h. Three days post infection, cells were trypsinized and GFP-positive cells were counted by flow cytometry on a BD FACSCalibur at the Tufts Laser Cytometry core facility. FlowJo (BD Biosciences) was used to analyze the data.

### 2.7. Measurement of Viral DNA Replication

6.8 × 10^4^ KSHV-infected iSLK.RTA cells were plated and treated with 1 μg/mL doxycycline to induce the lytic cycle. Four days post-induction, DNA from cells and supernatant was collected and purified using the DNeasy kit according to the manufacturer’s protocol. Viral DNA was measured as described above for virion DNA. Levels of the cellular gene CCR5 [26] were measured to normalize by total DNA extracted. Primers used for qPCR are listed in Table 1.

### 2.8. Virion Isolation

ORF42-Flag KSHV-infected iSLK.RTA cells were plated at a density of 3.06 × 10^6^ cells per dish on six 60-cm^2^ dishes. The lytic cycle was induced by addition of doxycycline (1 μg/mL). Six days post-induction the supernatant was collected and filtered to remove cells and debris. The supernatant was centrifuged at 80,000× g for 2 h at 4 °C through a 30% sucrose cushion in TNE buffer (50 mM Tris–HCl pH 7.4, 100 mM NaCl, 0.1 mM EDTA). The pelleted virions were resuspended in TNE buffer and underwent a second identical spin. The pelleted virions were then lysed, and proteins were isolated using RIPA buffer (50 mM Tris–HCl pH 7.4, 1% NP-40, 0.5% Na-deoxycholate, 0.1% SDS, 150 mM NaCl, 2 mM EDTA).

### 2.9. Protein Isolation and Western Blotting

The lytic cycle was induced in KSHV-infected iSLK.RTA cells by addition of doxycycline (1 μg/mL). Where indicated, the viral DNA polymerase inhibitor phosphonoacetic acid (PAA) was also added at the same time as doxycycline (day 0) at a concentration of 100 μg/mL to block viral DNA replication. Cells were lysed at the relevant days post-induction (indicated for each experiment in the figure and text) and proteins were isolated using RIPA buffer. Lysates were cleared by centrifugation and total protein concentrations were quantified by Bradford assay (Bio-rad). Protein lysates were separated by SDS-PAGE and transferred to PVDF membranes. Membranes were blocked in 5% milk in PBST (phosphate-buffered saline with 0.1% Tween-20 (Fisher)) and incubated with primary antibodies for 3 h at room temperature or overnight at 4 °C in PBST with 0.5% milk. Primary antibodies against the following proteins were used: KSHV ORF59, K8.1, ORF68 (all 1:10,000 dilution, gifts from Dr. Britt Glaunsinger [35,36]) KSHV ORF26, ORF33 and ORF52 (1:1000 dilution with the exception of ORF26, used at 1:5, gifts from Dr. Fanxiu Zhu [37,38]), KSHV ORF6 (1:500, gift from Dr. John Karijolich and Gary Hayward), LANA (1:1000 dilution in Tris-buffered saline with 0.1% Tween-20 (Fisher) and 0.5% milk, a gift of Dr. Patrick Moore [39]), Flag (1:500, Sigma), human β-actin (1:200, Santa Cruz Biotechnology), human β-tubulin (1:1000, Sigma), GFP (1:1000, Sigma). Secondary anti-rabbit and mouse (Southern Biotech, Birmingham, AL, USA) or anti-goat (Southern Biotech or Santa Cruz Biotechnology, Dallas, TX, USA) antibodies conjugated to horseradish peroxidase were used at a dilution of 1:5000 in PBST with 0.5% milk. Membranes were stained using Pierce ECL Western blotting substrate (Thermo Fisher) and imaged with a Syngene G:Box Chemi XT4 gel doc system. Images were quantified using GeneTools version 4.02.03.

### 2.10. RNA Isolation and RT-qPCR

Total RNA was extracted using a Quick-RNA MiniPrep kit (Zymo Research, Irvine, CA, USA) and treated with Turbo DNase (Ambion/Thermo Fisher) before reverse transcription. For reporter mRNA measurements, cDNA was prepared using an iScript cDNA Synthesis Kit (Bio-Rad) per the manufacturer’s protocol. For viral mRNA measurements, cDNA was prepared using AMV RT (Promega, Madison, WI, USA) per the manufacturer’s protocol, using a cocktail of primers targeted at KSHV genes and 18S ribosomal RNA (rRNA), in order to obtain strand-specific measurements. In both cases, human 18S rRNA levels were used as an internal standard to calculate relative mRNA levels. Real-time qPCR was performed using iTaq Universal SYBR Green Supermix (Bio-Rad) in a CFX Connect Real-time PCR Detection System (Bio-Rad). Primers used for qPCR are listed in Table 1. No template and no RT controls were included in each replicate. The CFX manager software was used to analyze the data.

### 2.11. Nuclear-Cytoplasmic Fractionation

To isolate cytoplasmic vs. nuclear fractions, we used the REAP fractionation method with minor modifications [40]. Cells were lysed in 0.1% NP40 in Dulbecco’s phosphate-buffered saline (DPBS). After collection of a whole-cell sample, the nuclear pellet was spun down. The supernatant was then collected as the cytoplasmic fraction. The nuclear pellet was washed in 0.1% NP40 in DPBS, and then lysed using RIPA buffer. Proteins from the fractions were then analyzed by Western blotting.

### 2.12. Metabolic Labeling

1.7 × 10^5^ KSHV-infected iSLK.RTA cells were treated with 1 μg/mL doxycycline to induce the lytic cycle. Five days post-induction the media was replaced with methionine-free media, and an hour later Click-iT^®^ L-azidohomoalaine (AHA, Thermo Fisher) was added to a final concentration of 0.1 μM. After 2 h, cells were fixed in 4% paraformaldehyde, washed with PBS, blocked with 1X Permawash (BD Biosciences) in PBS for 30 min or overnight, and incubated in 1 μM Dylight 650-Phosphine in 1X Permawash in PBS for 2 h to couple the fluorophore to AHA. After washing with PBS, AHA incorporation into proteins was measured by flow cytometry on a BD LSRII flow cytometer at the Tufts Laser Cytometry core facility. Flow cytometry data were analyzed using FlowJo version 10 to obtain the median fluorescence intensity (MFI) of cells.

### 2.13. Statistical Analysis

All statistical analysis was performed using GraphPad Prism version 8.0 for Mac OS X, (GraphPad Software, San Diego, CA, USA). Statistical significance was determined by Student’s t-test for individual comparisons or analysis of variance (ANOVA) followed by Dunnett’s, Tukey’s or Sidak’s multiple comparison test when multiple comparisons were required.

## 3. Results

### 3.1. Generation and Initial Characterization of KSHV ORF42^PTC^ Infected Cells

Annotation of the KSHV genome by Russo et al. in 1996 predicted the existence of a 31-kDa, 278-amino acid protein with homology to herpesvirus saimiri ORF42, which was also termed ORF42 [33]. Consistent with this original annotation, a comprehensive transcriptomic and Riboseq analysis of KSHV-infected cells also detected an mRNA and a translated open reading frame with start and stop codons corresponding to the original in silico annotation of the ORF42 gene [25]. To date, there has been no characterization of ORF42 or its homologs in other gamma-herpesviruses, including ORF42 from MHV68 and BBRF2 from Epstein Barr virus [41,42]. To study ORF42 and its function in the KSHV replication cycle, we inserted a premature termination codon at Serine 25 of ORF42 in the KSHV BAC16 clone using BAC recombineering (Figure 1A). We termed this BAC “ORF42^PTC^”. This mutation should abolish expression of full-length ORF42. Although other truncated forms starting at later in-frame methionines (for example Met 58) could still be produced, our results using a C-terminal Flag-tagged ORF42 encoded on the KSHV genome suggest additional isoforms are not produced at detectable levels (see Section 3.3). The sequence alteration we introduced generates a synonymous mutation in the coding sequence of ORF43 (at Valine 585), which overlaps with the 5’ end of the ORF42 coding sequence and is not expected to affect ORF43 expression and activity (Figure 1A). We also generated a revertant BAC, ORF42^REV^, in which the ORF42^PTC^ was mutated back to the wild-type sequence, in order to confirm that the recombineering had not compromised the integrity of the rest of the BAC. We checked the integrity of the viral BACs and their terminal repeats with restriction fragment length polymorphism analysis (Figure 1B). We also amplified the ORF42 region by PCR and sequenced it, thus confirming the presence or absence of the point mutation in the engineered BACs.

After engineering the mutated BACs, we generated cell lines latently and stably infected with wild-type (WT) KSHV, ORF42^PTC^ and ORF42^REV^ by transfecting the BACs in iSLK.RTA cells. iSLK.RTA cells are epithelial cells that express the KSHV master lytic regulator RTA from a doxycycline-inducible transgene integrated into the cellular genome [31]. RTA overexpression triggered by doxycycline treatment is sufficient to initiate the KSHV lytic cycle. We isolated clonal lines of iSLK.RTA cells infected with KSHV ORF42^PTC^, KSHV ORF42^REV^, or WT KSHV. We initially analyzed several WT KSHV-infected lines for virus production, and found variability among them, even though they were derived from the same BAC transfection. To conclusively determine that any observed phenotype really reflected the effect of the ORF42 mutation, rather than variability between cell lines, we decided to include two separate WT KSHV-infected cell lines as reference for most of our experiments.

The first readouts we tested were the levels of the key latent nuclear antigen (LANA) protein in latently infected cells, which is indicative of the establishment of a latent infection, and the expression of an early lytic gene, ORF57, after induction of the lytic cycle (i.e., doxycycline addition), which is indicative of the initial steps of reactivation from latency. The levels of LANA were similar in all lines (Figure 1C). Moreover, doxycycline treatment resulted in a dramatic increase in ORF57 mRNA levels in all infected cells (Figure 1D). Since the fold change in ORF57 levels is similar in all infected cells, we conclude that loss of full-length ORF42 did not compromise reactivation of the lytic cycle from latency, and that ORF42 is dispensable for KSHV reactivation from latency. Importantly, since WT KSHV and KSHV ORF42^PTC^ latently infected cells appear to reactivate similarly, we were able to use these cell lines to study the role of ORF42 during the rest of the replication cycle.

### 3.2. KSHV ORF42 is Required for Efficient Production of Virions

To determine whether ORF42 is required for viral replication, we measured infectious virus production from cells infected with the KSHV WT, ORF42^PTC^ and ORF42^REV^. We induced the lytic cycle in KSHV-infected cells by adding doxycycline, collected supernatant six days after induction and used it to infect naïve HEK293T cells. Because the KSHV BAC16 constitutively expresses GFP [31], the titer of infectious virions produced can be estimated from the percentage of GFP-positive infected HEK293T cells by flow cytometry, as previously done in other studies [43,44]. This measurement represents cell-free virus released from the cells into the supernatant. We found that cells infected with KSHV ORF42^PTC^ produced significantly fewer infectious virions than WT-infected cells (Figure 1E). In addition, virus production from cells infected with the ORF42^REV^ virus was similar to that from WT-infected cells, indicating that the replication defect was not due to problems with the virus backbone (Figure 1E). These results suggest that ORF42 plays an important role in viral replication. While we did not extensively analyze multiple ORF42^PTC^-infected lines, we found that other clonally isolated KSHV ORF42^PTC^-infected cells also produced lower titers of viruses. As mentioned above, it is formally possible that truncated forms of ORF42 are still expressed in the cells infected with KSHV ORF42^PTC^, as there are in-frame AUGs later in the open reading frame. However, the fact that the ORF42^PTC^ mutation reduces virus production indicates that full-length ORF42 is required for wild-type levels of viral replication.

Because we found variability in virus production from WT KSHV-infected clonal lines (Figure 1E), we wanted to test the same clonal latently infected line with and without ORF42 expression. In order to confirm the inhibitory effect of the ORF42^PTC^ mutation on viral replication, we examined the ability of full-length ORF42 express from a transgene to complement the KSHV ORF42^PTC^ virus and restore efficient viral production. To do so, we transduced KSHV ORF42^PTC^-infected cells with transgenes encoding untagged ORF42 or a Flag-tagged version of the protein. Rescue of the viral protein in trans is increasingly common in studies of KSHV mutants, to control for potential differences among the stable latently infected lines [45,46,47,48]. As a matching negative control, we transduced cells with a Flag tag-only empty vector, which controls for any effects of transduction and antibiotic (zeocin) selection for the transgene. Expression of the untagged ORF42 protein in trans rescued virus production from the ORF42^PTC^-infected cells, whereas transduction of the empty vector did not (Figure 2A). While there was no significant difference in virus production between ORF42^PTC^-infected cells and ORF42^PTC^-infected cells transduced with the empty vector, there was a trend toward slightly higher virus production from the empty vector transduced cells. This suggests that transduction or antibody selection may have a minor effect on induction of the lytic cycle. The fact that complementing ORF42 in trans rescues viral production confirms that the virus production defect observed in the ORF42^PTC^ virus is due to a loss of function effect of our ORF42 mutation. While untagged ORF42 significantly rescued infectious virus production, a C-terminally Flag-tagged ORF42 increased viral titers but did not fully rescue them, and the virus production remained significantly different from that of WT KSHV-infected cells (Figure 2A). This result suggests that the C-terminal tag may reduce ORF42 activity, similarly to what has been reported for the cytomegalovirus ORF42 homolog UL103 [14], or alter ORF42 production from the transgene. We also tested how expression of ORF42 in trans affected LANA expression in latently infected cells, as well as reactivation from latency by measuring the mRNA levels of the early gene ORF57 before and after induction of the lytic cycle. LANA protein levels during latency and ORF57 mRNA levels after lytic induction were similar in all KSHV WT and KSHV ORF42^PTC^-infected cell lines (Figure 2B,C, Table 2). ORF57 mRNA levels in latently infected cells appeared increased in the complemented lines, including cells transduced with the empty vector control, which may indicate that the transduction or antibody selection promoted some spontaneous reactivation (Figure 2C). However, the change did not reach statistical significance, and the basal levels of ORF57 in these lines remained ~100-fold lower than after lytic induction (Figure 2C). Because complementing ORF42 in trans allows us to test the same clonal latently infected line with and without ORF42 expression and reduces potential confounds, we used the ORF42 complemented cell line and the matching negative “empty” control to further characterize the effect of ORF42 in the viral replication cycle.

To test whether ORF42 was important for DNA replication, we examined the levels of viral DNA in latent and lytic infection. We found that the levels of viral genomes were similar in cells infected with KSHV WT and ORF42^PTC^ and in the complemented lines during latency (Figure 2D). Lytic reactivation led to a similar increase in viral DNA levels in all the lines (Figure 2E). While the fold increase in DNA levels was variable, none of the differences reached statistical significance. In particular, DNA replication was not reduced in ORF42^PTC^-infected cells, suggesting that loss of full-length ORF42 does not impact DNA replication (Figure 2E). There was an apparent increase in DNA replication in all transgene-rescued cells relative to the KSHV WT-infected cells, although the differences were not statistically significant. This apparent increase may be due to increased reactivation due to zeocin treatment or transduction. Nonetheless, overall the analysis of viral DNA replication suggests that the reduction in infectious virions produced by ORF42^PTC^-infected cells is not due to the presence of fewer replicated genomes to be packaged.

Because ORF42 homologs are tegument proteins that could be involved in early stages of de novo infection, loss of full-length ORF42 could reduce particle production, particle infectivity, or both. To distinguish among these possibilities, we estimated the released viral particles by measuring DNase-resistant viral DNA genomes in the supernatant with qPCR. We found that the changes in viral particles mirrored those in infectious titers: KSHV ORF42^PTC^-infected cells produced lower levels of cell-free viral particles, estimated from viral DNA in the supernatant, and complementation of ORF42 expression in trans abrogated this defect (Figure 2F). While transduction of the cells with the empty vector control and/or selection of the transduced cells with zeocin also induced higher levels of viral particles (i.e., genomes in the supernatant), this was still substantially and significantly lower than the virus production in cells infected with WT KSHV, or KSHV ORF42^PTC^ in the presence of the ORF42 rescuing construct (Figure 2F). These results suggest that ORF42 affects the formation and/or release of viral particles. However, we cannot exclude that loss of full-length ORF42 may also impact infectivity in other target cell types, for example more relevant cell types such as endothelial and B cells, especially since we found that ORF42 is present in virions (see Section 3.3).

It has been proposed that ORF42 homologs of other herpesviruses may act specifically in virion release [12,13]. The most convincing evidence is that in the absence of PRV UL7, nucleocapsid formation, nuclear egress and budding of capsids into the Golgi appeared normal by electron microscopy, but the release of virions from the cell was severely inhibited [13]. Moreover, UL7 mutations had a minimal effect on the accumulation of intracellular (cell-associated) virus levels [13]. To investigate how particle release was affected by ORF42, we isolated cell-associated virions from KSHV-infected cells and measured intracellular viral titers. We found that ORF42 was clearly required for efficient production of intracellular virions prior to release from the cell (Figure 2G). Based on these results, it is unclear whether virion release is affected by ORF42 mutations. Minimally these results indicate that particle release from the cells is not the only process that is compromised in the absence of full-length ORF42.

Collectively, our initial characterization of the ORF42 mutant KSHV reveals that full-length ORF42 is essential for wild-type levels of virion production but does not affect reactivation of the lytic cycle or viral DNA replication.

### 3.3. ORF42 Is a Late Cytoplasmic Protein

To characterize the localization and the temporal expression of ORF42 in the context of the lytic replication cycle, we also made a KSHV BAC16 variant that expresses a C-terminally Flag-tagged version of ORF42 from the viral genome (KSHV ORF42-Flag) (Figure 3A). We chose a C-terminal tag, because we deemed an N-terminal tag too disruptive, given the overlap between the promoter and 5’ end of the ORF42 coding region and ORF43 (Figure 1A). To accommodate a C-terminal tag, we had to duplicate the 3’ end of the ORF42 coding sequence after the Flag tag stop codon (Figure 3A, Appendix A), because the ORF41 transcription termination and polyadenylation sequences and a portion of the ORF41 coding region overlap with the 3’ end of the ORF42 coding sequence (in the complementary strand, Figure 1A). This insertion also provides the polyadenylation signal for ORF42, which overlaps the ORF42 stop codon (Figure 3A). We checked integrity of the BAC with restriction fragment length polymorphism analysis (Figure 3B). PCR analysis of the BAC in the ORF42-ORF43 region confirmed the presence of a ~150 bp insertion, which matches the expected 144 bp size of the duplicated sequence and Flag tag (Figure 3C). The insertion was also verified by Sanger sequencing. We then generated KSHV-infected cells by transfecting the KSHV ORF42-Flag BAC in iSLK.RTA cells. Consistent with the in silico annotation of the ORF42 locus [33] and the Riboseq results from Arias et al. [25], addition of the Flag tag at the 3’ end of the ORF42 coding sequence led to the detection of a ~35 kDa protein in lysates from cells infected with KSHV ORF42-Flag but not WT KSHV using an anti-Flag antibody (Figure 3D). No additional products were observed, confirming that the currently annotated ORF42 coding sequence corresponds to the main isoform of ORF42 expressed in iSLK cells (Figure 3D). This is important to note, because the organization of the ORF42 genomic locus and the presence of in-frame methionines at position 59 and 62 of ORF42 would potentially allow for the production of additional shorter isoforms of ORF42, for example from the ORF43/ORF42 bicistronic transcript that is used to produce ORF43 [25]. These isoforms would be ~28 kDa (including the Flag tag). However, the absence of additional bands in the Western blot suggests that shorter isoforms are not produced at any appreciable rate, at least in the absence of any mutations in the ORF42 coding sequence. The KSHV ORF42-Flag virus produced infectious virions after induction with doxycycline (Figure 3E), but the titer was less than that for WT KSHV, suggesting that C-terminal tagged ORF42 was less active, as we also saw in Figure 2A. Nonetheless, we used this system to examine the expression kinetics and localization of the Flag-tagged ORF42.

We found that virus-encoded ORF42-Flag was expressed with the kinetics of a late lytic protein (Figure 3D,F). It was detected in lytically reactivated cells at the same time as the late proteins K8.1 and ORF26, while the early protein ORF59 was expressed earlier (Figure 3D,F). In addition, ORF42 production was sensitive to treatment with phosphonoacetic acid (PAA), an inhibitor of viral DNA replication, which is a hallmark of a canonical late protein (Figure 3F). These data agree with results from a previous transcriptome-wide analysis of KSHV mRNAs in the presence of the viral DNA polymerase inhibitor cidofovir [49]. The late expression is also consistent with our finding that ORF42 is not required for reactivation and viral DNA replication (Figure 1D and Figure 2C–E).

In terms of subcellular localization, we found that virus-encoded ORF42-Flag was localized almost exclusively to the cytoplasm, as indicated by Western blot analysis of fractionated cell lysates (Figure 3G). In addition, we detected ORF42 in virions isolated by ultracentrifugation (Figure 3H). This result indicates that ORF42 is indeed a component of the virion and probably of the tegument, the layer of proteins between the herpesviral capsid and envelope, like its homologs in alpha- and beta-herpesviruses [9,10,11,13]. It is unlikely that the ORF42 staining in the isolated virions is due to cellular debris, because we did not detect the viral DNA polymerase processivity factor ORF59, a protein that is not usually found in the virions (Figure 3H). We conclude from these experiments that KSHV ORF42 is a late lytic and cytoplasmic protein that is incorporated into the viral tegument.

### 3.4. ORF42 Is Required for Wild-Type Levels of Some Viral Proteins

A hallmark of the progression through the KSHV replication cycle is the temporally regulated expression of viral genes. In particular, early genes are expressed prior to DNA replication, while late gene expression requires DNA replication. Therefore, we examined the expression of KSHV proteins from different stages of the replication cycle at Day 4 and 6 post-induction. These time points were selected because we were interested in analyzing gene expression prior to and during production of infectious virions, and we have seen robust production of virions starting at Day 5. We measured the levels of multiple early (Figure 4A) and late proteins (Figure 4B). As antibodies against KSHV proteins are limited, these viral proteins were chosen based on the availability of antibodies, rather than specific functions. At Day 4 we found that the levels of the early proteins ORF59, ORF6 and ORF68, which are all involved in DNA replication, were slightly decreased in the KSHV ORF42^PTC^-infected cells compared to the WT KSHV-infected cells (Figure 4A, Table 2). In contrast, the ORF42^PTC^ mutation substantially reduced the levels of several late proteins: the glycoprotein K8.1, the capsid protein ORF26, and the tegument proteins ORF33 (Figure 4B, Table 2). We also tested the expression of another late tegument protein, ORF52, but found that there was no consistent defect in ORF52 expression in the KSHV ORF42^PTC^-infected cells (Figure 4B, Table 2). At Day 6, we again found that, while almost all the proteins tested were produced at lower levels in cells infected with KSHV ORF42^PTC^, there was a more dramatic effect on the late proteins K8.1, ORF26 and ORF33 (Figure 4A,B, Table 2). Complementation of ORF42 in trans restored high levels of all viral proteins (Figure 4A,B, Table 2). Transduction of the empty vector increased levels of some proteins slightly, which may be due to the slight induction of lytic reactivation and DNA replication seen in Figure 2. However, levels of all proteins tested (except ORF52) were on average lower in empty vector transduced cells than ORF42-complemented cells (Figure 4A,B, Table 2). This confirms that the reduced protein levels were due to loss of full-length ORF42 expression (Figure 4A,B, Table 2). These results indicate that full-length ORF42 was required to produce at least some of the viral proteins at wild-type levels. The change in protein levels was unexpected because no change in viral protein levels has been reported for other herpesviruses carrying mutations in the ORF42 homologs [12,13]. Therefore, this result suggests a potential additional function for ORF42 in KSHV. Based on our characterization of ORF57 induction and viral DNA replication in the mutant virus (Figure 1D and Figure 2C–E), we can also exclude that the alteration in protein levels is a secondary effect of early defects in reactivation and viral DNA replication. Therefore, we conclude that ORF42 plays a role in promoting viral gene expression, which in turn may affect virion production. The stronger effect on late proteins may be linked to the fact that ORF42 itself is a late protein and that the early proteins normally start to accumulate before ORF42 is made.

### 3.5. ORF42 Activity May Potentiate Gene Expression at a Post-Transcriptional Level

To determine whether ORF42 regulated the expression of viral proteins at the level of RNA or protein production, we measured RNA and protein in parallel for K8.1 and ORF26, the two genes whose expression was most clearly affected by the presence of ORF42. At Day 4 post-induction, we found that the protein levels of K8.1 were significantly reduced in the absence of ORF42, while there was no significant difference in the RNA levels (Figure 5A). At Day 6, we observed a significant reduction in both RNA and protein levels for K8.1, but the change in RNA levels was smaller in magnitude (Figure 5B). For ORF26, we found that there was an apparent ORF42-dependent reduction in protein but not mRNA levels at Day 4, although it did not reach statistical significance (Figure 5C). At Day 6, there was a statistically significant decrease in ORF26 protein levels, but not mRNA levels, between KSHV WT and ORF42^PTC^-infected cells (Figure 5D).

Therefore, while there was some difference between the two targets, the overall effects followed a similar trend: only protein levels were decreased in the absence of ORF42 at Day 4, whereas at Day 6 RNA levels were also reduced, but the magnitude of the change was smaller.

To extend these observations and test whether ORF42 affects protein production, we overexpressed ORF42 together with a reporter construct mimicking the mRNA for ORF26 (including untranslated regions), but driven by the cytomegalovirus (CMV) promoter. In this case, ORF26 transcription was not controlled by the native KSHV promoter, which meant that its RNA levels were not controlled by KSHV factors whose expression may be influenced by ORF42. We found that overexpression of ORF42 increased protein levels for ORF26, but had no effect on its RNA levels (Figure 5E). Because ORF42 appeared to have an effect on the levels of the ORF26 protein in the absence of transcriptional regulation, these results suggest that ORF42 may promote protein rather than RNA production.

As an additional test of ORF42-dependent control of protein production in KSHV-infected cells, we measured the effect of ORF42 on total nascent protein production. We treated cells with a Click-able methionine analogue L-azidohomoalanine (AHA), which is incorporated into nascent proteins. Because AHA can be conjugated to a fluorescent dye, AHA incorporation into newly synthesized proteins can easily be measured by flow cytometry. This assay not only provides a more global picture of protein production in the cell, but also specifically measures new protein synthesis. We tested KSHV ORF42^PTC^-infected cells rescued with WT ORF42 or empty vector, which represent the exact same infected cellular line subjected to the same antibiotic selection in the presence or absence of full-length ORF42. The median fluorescence intensity of KSHV lytically-infected cells was 10% lower in the absence of ORF42, suggesting that in the presence of ORF42 total protein production is elevated by 10% (Figure 6A,B). This difference was reproducible and statistically significant (Figure 6B). We note that it is likely that not all KSHV-infected cells are lytically reactivating in this assay, and that we may thus be underestimating the effect of ORF42 on translation in infected cells. Nevertheless, this small but consistent change indicates that the presence of ORF42 promoted the synthesis of new proteins in infected cells.

## 4. Discussion

The role of many KSHV proteins in the viral replication cycle remains uncharacterized, particularly if these proteins do not have well-studied homologs in other herpesviruses nor resemble cellular proteins of known function. Here we have started characterizing the role of a protein of unknown function, ORF42, in viral replication using KSHV BAC16 recombinants harboring either a premature termination codon in ORF42 or a Flag tag at the C-terminus of ORF42. We found that ORF42 is a late cytoplasmic and virion protein that is necessary for efficient replication of KSHV. In the absence of full-length ORF42, KSHV replication is reduced more than 50-fold. Consistent with the fact that ORF42 expression itself is dependent on DNA replication, ORF42 is not required for reactivation of the lytic cycle or viral DNA replication. However, we found that ORF42 is required for efficient accumulation of at least some viral proteins. Our analysis of select viral proteins suggests that the activity of ORF42 may be promoting protein production, rather than transcription of viral genes. While ORF42 homologs in other herpesviruses are also required for wild-type levels of viral replication, they have generally not been reported to affect viral gene expression, particularly at the level of protein production. This suggests that KSHV ORF42 may have additional or alternative functions compared to its alpha and beta-herpesviral homologs.

Because we introduced an early nonsense mutation in the ORF42^PTC^ mutant virus and we do not have an ORF42 antibody, we cannot exclude the possibility that truncated shorter forms of the protein are still produced in KSHV ORF42^PTC^-infected cells from later in-frame methionines. Since an ORF43/ORF42 bicistronic transcript is produced to translate ORF43 [25], shorter isoforms of ORF42 could be produced from the bicistronic transcript, through re-initiation at ORF42 AUG codons after the ORF43 stop codon. Such products would still be produced in the ORF42^PTC^ virus. However, we consider this possibility unlikely, because we detected a single Flag-tagged product in KSHV ORF42-Flag infected cell lysates (Figure 3C). This product is of the size expected based on the original in silico annotation [33] and previous ribosome profiling data [25]. The ribosome profiling results also did not report other initiation codons inside ORF42 [25]. Therefore, shorter isoforms of ORF42 do not appear to be produced at any appreciable levels in the wild-type sequence context. Moreover, since the ORF42^PTC^ mutation is located upstream of the ORF43 stop codon, there is no reason to believe it would have any stimulatory effect on translation from the bicistronic construct. In the context of the ORF42^PTC^ virus, it also remains possible that in-frame AUG codons later in the ORF42 coding sequence could be used to produce N-terminally truncated forms of the protein from the ORF42 monocistronic transcript. This process would require translation re-initiation, perhaps with the short remaining ORF42 aa 1–24 coding sequence acting as a translation-promoting short upstream ORF (uORF) [50]. However, the early nonsense mutation we inserted in ORF42 reduced virus production by more than one log (Figure 1 and Figure 2). Therefore, even if any truncated forms of ORF42 are produced by KSHV ORF42^PTC^, these forms are not sufficient to provide the activity necessary for efficient viral replication. Moreover, we are able to rescue the defects we see in the ORF42^PTC^ mutant virus by expressing a full-length version of ORF42 from a transgene. This result suggests that the defects we see in KSHV ORF42^PTC^ are due to the lack of ORF42 function, rather than aberrant dominant negative effects of ORF42 truncated forms.

Prior to this study, ORF42 was frequently annotated simply as a tegument protein, because its alpha- and beta-herpesvirus homologs are tegument proteins [9,10,11]. Indeed, we found that ORF42 accumulates in virions, and presumably in the tegument (Figure 3H), even though most mass-spectrometry studies of gamma-herpesviruses were unable to detect it [17,18,19,20,21,22]. We speculate that only a few molecules of ORF42 may be in the virions. Like many tegument proteins, including KSHV ORF33, ORF38, ORF45, and ORF52 [37,48,51], the ORF42 homologs in alpha- and beta-herpesviruses UL7 and UL103 have been implicated in virion assembly and release, although the specific role of the homologs remains unknown. Interestingly, most prior studies of UL7 and UL103 did not find any effect of these proteins on earlier steps of viral replication, including viral protein levels. In contrast, our results suggest a role for ORF42 in maintaining expression of at least some late proteins. While phylogenetic analysis reported in the Pfam database suggests that alpha-herpesvirus UL7, HCMV UL103 and KSHV ORF42 belong to the same protein superfamily and thus are likely true homologs [8], the sequence identity between the proteins is low (<20%). Therefore, our results may be indicative of the acquisition of a separate or alternative function for ORF42 in the KSHV replication cycle. That said, while the reduction in virion and tegument protein levels in the absence of ORF42 may explain the reduced viral production, there is a discrepancy in the magnitude of the reduction in viral proteins vs. virus production. The change in protein accumulation was less than 10-fold (Figure 4 and Figure 5, Table 2), while the defect in virus production was at least 50-fold (Figure 1E and Figure 2A,F). Because of the reduced accumulation of virion and tegument proteins in cells infected with KSHV ORF42^PTC^, we cannot accurately assess a separate role in virion assembly. While it is possible that changing the stoichiometry of viral proteins could have disproportionate effects on virus assembly, the discrepancy may point to multiple functions for ORF42 in the KSHV replication cycle, including a separate function in virus assembly similar to that of its homologs. It will also be interesting to test whether ORF42 homologs in other gamma-herpesviruses similarly regulate viral gene expression and/or have a dual function.

We note that the iSLK cells do not represent the relevant cell type for KSHV infection in vivo, as they are of epithelial origin [52]. Endothelial and B cells are more relevant cell types for KSHV infection. Kaposi’s sarcoma lesions are composed of infected endothelial cells, and infected B cells are thought to be the reservoir of KSHV latent infection in patients [53]. B cell transformation also leads to the development of KSHV-associated lymphomas, such as primary effusion lymphoma [2]. Unfortunately, currently there are no systems to reconstitute and study KSHV mutants in the context of the lytic cycle in endothelial and B cells. Therefore, the iSLK-BAC16 system remains the best system available to study the function of KSHV proteins in lytic replication using mutated KSHV strains, as well as the control of lytic viral gene expression. Nonetheless, viral products can have cell-type specific roles, as previously demonstrated for HCMV and HSV-1 proteins [54,55,56,57]. Therefore, as the recombinant systems for studying KSHV are further developed, it will be interesting to study how ORF42 contributes to infection in more relevant cells, such as B cells and endothelial cells.

We propose that ORF42 has a function in promoting viral protein production because loss of full-length ORF42 reduced the levels of viral proteins (Figure 4 and Figure 5, Table 2). As transcriptional induction of early genes and viral DNA replication were normal in KSHV ORF42^PTC^ infected cells (Figure 1C and Figure 2C–E), the change in viral protein levels is unlikely to be a secondary effect of earlier problems in the replication cycle. We favor a post-transcriptional effect of ORF42 activity on gene expression, because loss of full-length ORF42 had only modest and delayed effects on RNA levels for the two late genes we tested (Figure 5A–D). Moreover, overexpression of ORF42 alone could increase protein expression of an ORF26 reporter expressed from a non-KSHV promoter, although it had no effect on its RNA levels (Figure 5E). Lastly, ORF42 appears to promote increased levels of total protein production during infection, as measured by metabolic labeling (Figure 6). If our conclusion is correct, the changes in viral RNA levels could be a secondary effect of reduced accumulation of viral proteins. Alternatively, since we detect ORF42-dependent changes in RNA levels at Day 6, ORF42 may affect multiple stages of gene expression, including transcription. We also do not currently know whether ORF42 has a direct effect on gene expression, for example through interactions with protein production machinery, or activates/inactivates a signaling pathway that regulates translation, similarly to the KSHV protein ORF45 [58]. Further work is needed to distinguish between these two possibilities. We also found that ORF42 had a greater effect on the expression of specific late proteins, particularly at Day 4 (Figure 4, Table 2). Because ORF42 is itself expressed with late kinetics, the easiest explanation is that early proteins have already accumulated to some level by the time ORF42 is expressed, and thus the impact of ORF42 activity on their levels is lower. However, we cannot exclude the possibility that late protein translation in general or production of specific KSHV virion proteins requires additional cellular and/or viral factors that are dispensable for early protein expression, as is the case for late gene transcription [59]. It will also be interesting to determine whether ORF42 has any effect on cellular protein production, since we observed an ORF42-dependent change in total protein synthesis.

How would having a KSHV-encoded factor like ORF42 that boosts accumulation of proteins later in the replication cycle benefit the virus? Translation can be inhibited by stress and innate immune responses [60], and levels of KSHV mRNAs are also likely affected by RNA degradation by the host shutoff RNase SOX [29]. Therefore, the activity of viral proteins that stimulate gene expression at a post-transcriptional stage could be important to maintain abundant viral late protein production. It may also be advantageous to boost viral gene expression only in the later stages of the replication cycle, to promote virus formation without exposing the virus to recognition by the immune system too early in replication. Further work on the mechanism of stimulation of gene expression by ORF42 will shed light on how the virus utilizes this protein to benefit the viral replication cycle, and allow us to expand our analysis to the functions of ORF42 homologs in other related viruses.

## Figures and Tables

**Figure 1 viruses-11-00711-f001:**
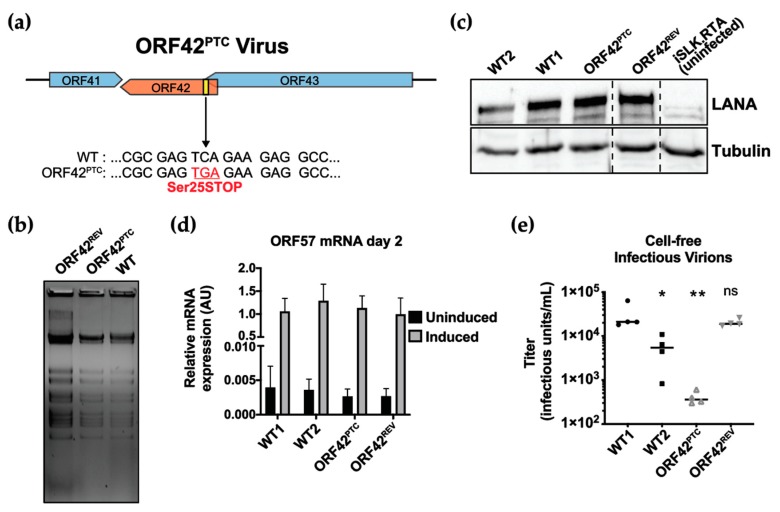
A premature termination codon in Kaposi’s sarcoma-associated herpesvirus (KSHV) ORF42 does not impair reactivation from latency, but attenuate viral replication. (**a**) Schematic diagram of the genomic locus surrounding KSHV ORF42 carrying a premature termination codon (ORF42^PTC^), including the location of the nonsense mutation at serine 25. (**b**) Wild-type (WT), ORF42^PTC^ and ORF42 revertant (ORF42^REV^) BACs were digested with the NheI restriction enzyme and resolved on a 0.4% agarose gel stained with ethidium bromide. (**c**) Protein lysates were collected from the indicated latently KSHV-infected iSLK.RTA cells, as well as uninfected iSLK.RTA cells. The latent protein LANA and the cellular protein β-tubulin were detected by Western blotting. (**d,e**) The lytic cycle was induced in the KSHV-infected iSLK.RTA cells by addition of doxycycline (1 μg/mL) to the media. In this and other figures, WT1 and WT2 refer to two clonal lines infected with wild-type KSHV. (**d**) Cells were treated with doxycycline (induced) or left untreated (uninduced). mRNA was collected 2 days after induction and RT-qPCR was used to measure the levels of the viral mRNA ORF57 and cellular 18S rRNA. ORF57 levels are reported after normalization to 18S. All differences between induced and uninduced samples were statistically significant using ANOVA followed by Sidak’s corrected multiple comparison test (*p* < 0.0001), but there was no significant difference between ORF57 levels in lytic (induced) cells among the strains (*p* > 0.05, ANOVA followed by Sidak’s corrected multiple comparison test). Bar graph represents mean ± standard deviation. *n* ≥ 4. (**e**) The supernatant from lytically reactivated cells was collected six days post-induction and used to infect HEK293T target cells. The percentage of GFP-positive target cells was measured by flow cytometry and used to calculate infectious units based on the Poisson distribution. Median titers are indicated. *n* ≥ 4. ns, *, ** = *p* > 0.05, < 0.05, and < 0.01, respectively. One-way ANOVA followed by Dunnet’s multiple comparison test vs. WT1.

**Figure 2 viruses-11-00711-f002:**
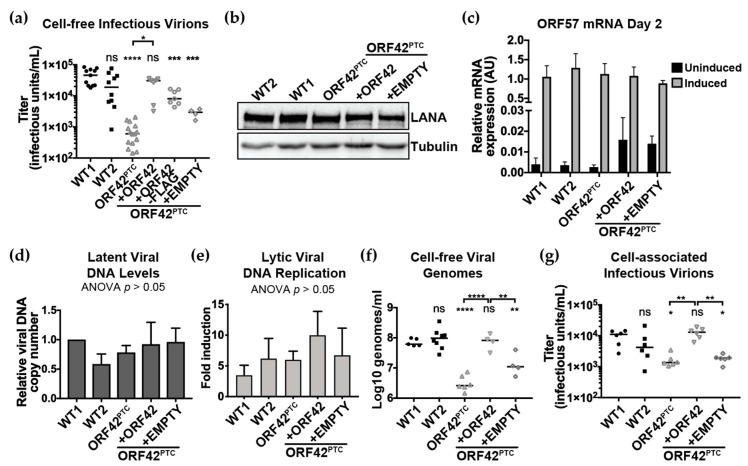
KSHV ORF42^PTC^ mutations reduce virion formation. The lytic cycle was induced in the indicated infected iSLK.RTA cells by addition of doxycycline (1 μg/mL) to the media. In addition to WT1, WT2 and ORF42^PTC^, in this and other figures the following labels are used: +ORF42-Flag: KSHV ORF42^PTC^-infected cells transduced with C-terminally Flag-tagged ORF42; +ORF42: KSHV ORF42^PTC^-infected cells transduced with untagged ORF42; +EMPTY: KSHV ORF42^PTC^-infected cells transduced with empty vector. (**a**) The supernatant from lytically reactivated cells was collected six days after induction and used to infect HEK293T target cells to estimate levels of cell-free infectious virions. Infectious units were calculated from the percentage of GFP-positive target cells measured by flow cytometry. (**b**) Protein lysates were collected from the indicated latently KSHV-infected iSLK.RTA cells. The latent protein LANA and the cellular protein β-tubulin were detected by Western blotting. (**c**) Cells were treated with doxycycline (induced) or left untreated (uninduced). mRNA was collected 2 days after induction and RT-qPCR was used to measure the levels of the viral mRNA ORF57 and cellular 18S rRNA. ORF57 levels are reported after normalization to 18S. All differences between induced and uninduced were statistically significant using ANOVA followed by Sidak’s corrected multiple comparison test (*p* < 0.0001), but there was no significant difference between ORF57 levels in lytic (induced) cells among the strains (*p* > 0.05, ANOVA followed by Sidak’s corrected multiple comparison test). (**d**–**e**) Total DNA was collected prior to induction and four days post-induction. qPCR was used to quantify levels of the viral gene LANA and the cellular gene CCR5. For (**d**), latent viral copy numbers are plotted relative to WT1 after normalization to CCR5. For (**e**), the fold increase in DNA levels after induction was calculated after normalization to CCR5 levels. For panels d and e, there is no statistically significant differences between any of the conditions, ANOVA *p* > 0.05. (**f**) The supernatant from lytically reactivated cells was collected six days after induction and qPCR was used to quantify viral DNA in supernatant using primers against KSHV LANA. This should estimate viral particle levels, because the samples were treated with DNase to remove unencapsulated DNA prior to DNA isolation. (**g**) Cell-associated (i.e., intracellular) virions were isolated and used to infect HEK293T target cells. Infection was quantified by determining the percentage of GFP-positive target cells by flow cytometry. For all panels, n ≥ 3. Bar graphs represent mean ± standard deviation. In the other graphs, lines indicate median titers. For panels a, f and g: ns, *,**,**** = *p* > 0.05, or < 0.05, < 0.01, and < 0.0001 respectively, ANOVA followed by Sidak’s multiple comparison test. The *p* values are either for comparisons to WT1 (where no bracket is present) or for the comparison indicated by the bracket. The WT2 vs. ORF42^PTC^ comparison for panels a and f is also significantly different, *p* < 0.01, ANOVA followed by Sidak’s multiple comparison test.

**Figure 3 viruses-11-00711-f003:**
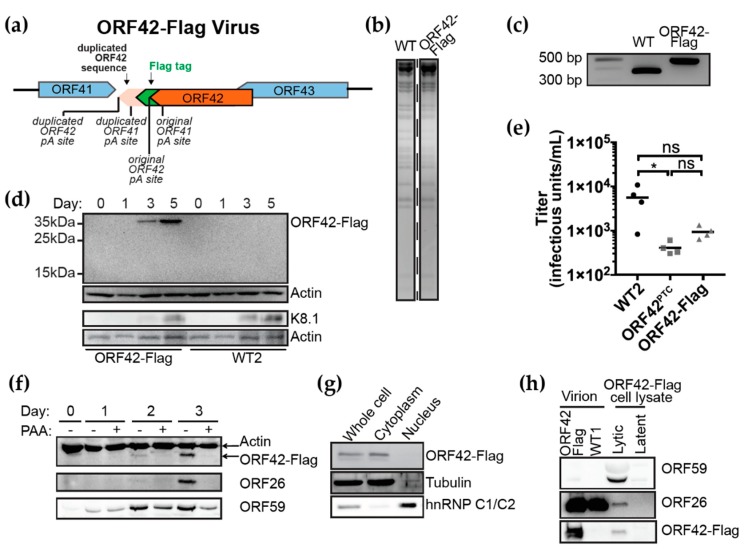
ORF42 is a late cytoplasmic protein. (**a**) Schematic diagram of the KSHV ORF42-Flag genome structure, indicating the C-terminal Flag tag position and the partial duplication of ORF42 sequences, including the original and duplicated ORF41 and ORF42 polyadenylation (pA) sites. (**b**) WT and ORF42-Flag BACs were digested with the KpnI restriction enzyme and resolved on a 0.4% agarose gel. (**c**) The modified region of the ORF42-Flag BAC16 was PCR amplified and resolved on a 1.5% agarose gel to confirm the presence of the insertion. (**d**) Protein lysates were collected at the indicated time points (day = days post-induction) from cells infected with WT KSHV or KSHV ORF42-Flag infected cells. ORF42 was detected using anti-Flag antibodies. Actin serves as a loading control and K8.1 as a control for lytic reactivation. (**e**) Supernatant was collected 6 days post-induction from cells infected with KSHV WT, ORF42^PTC^ or ORF42-Flag and used to infect HEK293T target cells. Infectious titers were calculated from the percentage of GFP-positive target cells determined by flow cytometry. Lines indicate median titers. ns, * = *p* > 0.05 and < 0.05, respectively. ANOVA followed by Tukey’s multiple comparison test. (**f**) Protein lysates were collected at the indicated time points from cells infected with KSHV ORF42-Flag and treated with doxycycline and the viral DNA polymerase inhibitor phosphonoacetic acid (PAA, 100 μg/mL). They were then analyzed by Western blot. Antibodies detecting ORF42-Flag, actin as a loading control, and ORF59 and ORF26 as representative early and late viral proteins were used. (**g**) Cells infected with KSHV ORF42-Flag were fractionated into cytoplasmic and nuclear fractions at Day 4 post-induction. Lysates were stained for ORF42-Flag, and tubulin and hnRNP C1/C2 as cytoplasmic and nuclear loading controls, respectively. (**h**) Six days post-induction, virions were isolated from the supernatant of cells infected with WT KSHV or KSHV ORF42-Flag and lysed to collect protein. Lysates from latent and lytic KSHV ORF42-Flag-infected cells are shown as staining controls. Western blotting was used to detect ORF42-Flag, ORF59 (polymerase processivity factor) and ORF26 (capsid protein). All blots are representative of ≥3 biological replicates.

**Figure 4 viruses-11-00711-f004:**
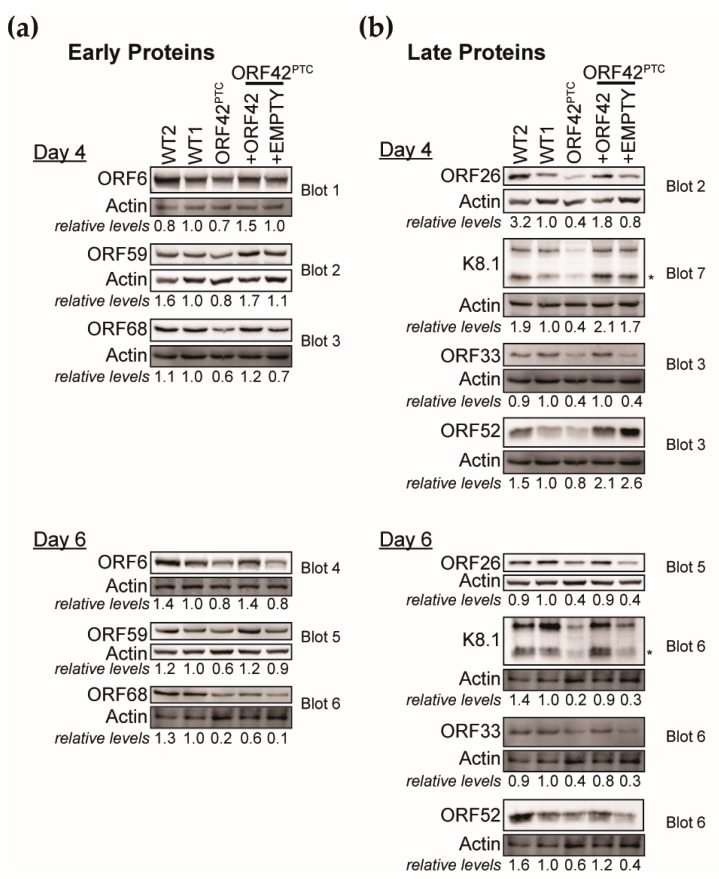
Loss of full-length ORF42 impairs the accumulation of viral proteins. The lytic cycle was induced in the KSHV-infected iSLK.RTA cells by addition of doxycycline (1 μg/mL) to the media. Protein was collected four and six days post-induction. Western blotting was used to measure the levels of the indicated early (**a**) and late (**b**) viral proteins. Expression is reported normalized to actin and relative to the WT1 line for the blot shown. All blots shown are from the same biological replicate and are representative of at least 3 biological replicates. Table 2 includes the average and standard deviation values for all the replicates. Several proteins were measured on the same blot and normalized to the same actin blot, as indicated by the blot number. For the glycoprotein K8.1, we used the band corresponding to the size predicted from the sequence for quantification (marked by the asterisk). Other bands likely represent glycosylated forms of K8.1.

**Figure 5 viruses-11-00711-f005:**
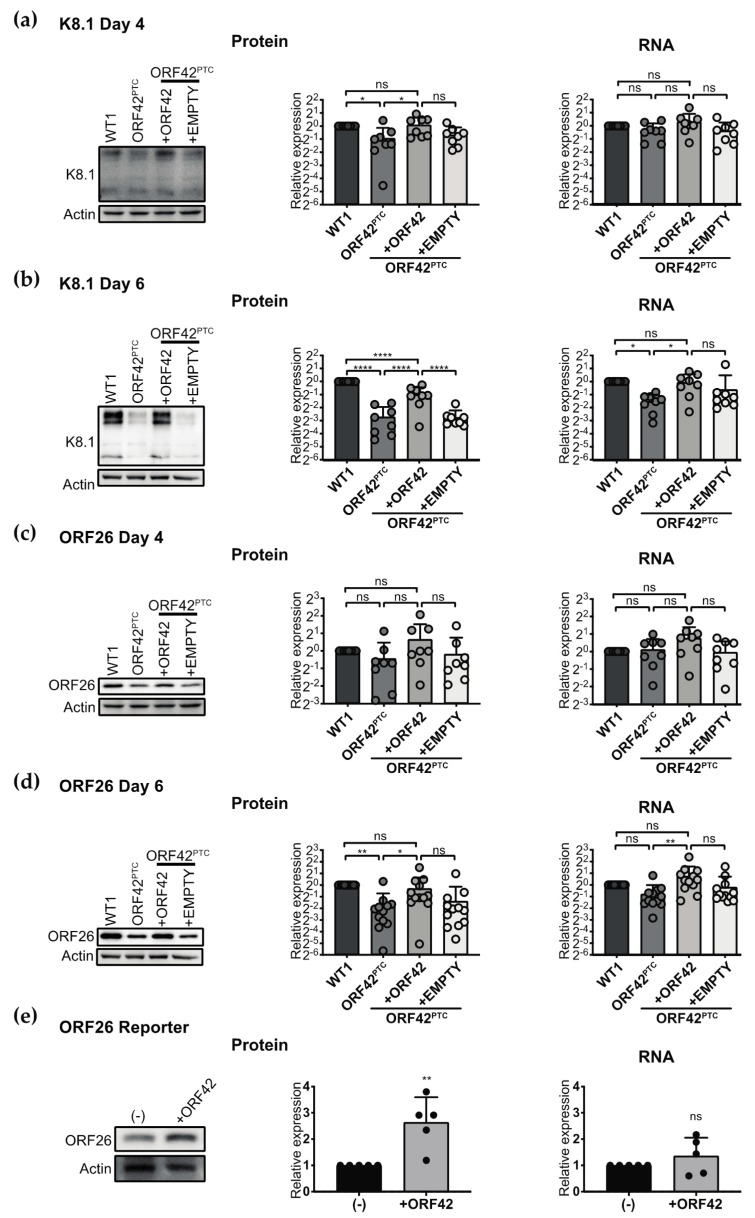
ORF42 may regulate gene expression post-transcriptionally. (**a**–**d**) The lytic cycle was induced in the KSHV-infected iSLK.RTA cells by addition of doxycycline (1 μg/mL) to the media. Protein and mRNA were collected four (**a**,**c**) and six (**b,d**) days post-induction from the same biological replicate. The levels of K8.1 (**a**,**b**) and ORF26 (**c**,**d**) proteins and mRNAs are reported relative to WT1. *n* ≥ 8. ns, *,**,**** = *p* > 0.05, < 0.05, < 0.01, < 0.0001 (ANOVA followed by Sidak’s corrected multiple comparison test). (**e**) A construct expressing the ORF26 coding region and UTRs was co-transfected with either ORF42 (untagged) or empty vector. Protein and RNA were collected 24 h later. *n* = 5; ns, ** = *p* > 0.05 or < 0.01, respectively (Student’s *t*-test). In all panels, levels of viral mRNAs and of cellular 18S rRNA were measured by RT-qPCR. Viral mRNA levels were then normalized to 18S rRNA levels and are shown relative to WT1 (**a**–**d**) or vector transfected cells (**e**). Protein levels were measured by Western blotting and normalized to actin, and are also shown relative to WT1 (**a**–**d**) or vector transfected cells (**e**). Representative Western blots are included. Bar graphs represent mean values ± standard deviation and individual measurements are marked.

**Figure 6 viruses-11-00711-f006:**
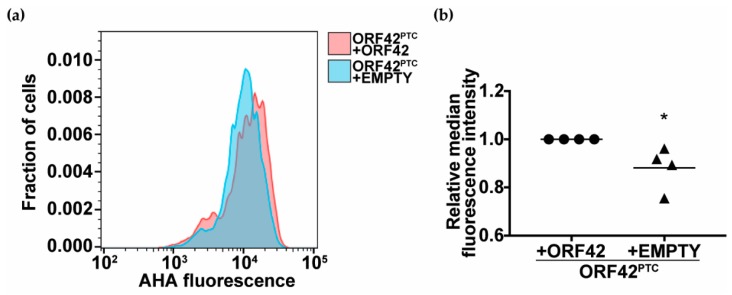
ORF42 affects total protein synthesis. Protein synthesis rates were measured using AHA-based metabolic labeling. The lytic cycle was induced in the indicated KSHV-infected iSLK.RTA cells by addition of doxycycline (1 μg/mL) to the media. Five days after lytic cycle induction, nascent proteins were labeled using AHA coupled to a fluorescent dye. The fluorescence intensity of AHA-treated cells was measured by flow cytometry. The distribution of fluorescence in the cells is plotted in (**a**) for a representative replicate. In (**b**) the median fluorescence intensity of the cell population is plotted relative to KSHV ORF42^PTC^-infected cells rescued with untagged ORF42. *n* = 4. * = *p* < 0.05 (Student’s *t*-test).

**Table 1 viruses-11-00711-t001:** Primers.

Construct	Sense	Purpose	Sequence
pJP1-ORF42-Flag	F	Amplify ORF42-Flag and add AgeI site for restriction enzyme-based cloning	CGCCACCGGTATGTCCCTGGAAAGGGCCCTG
pJP1-ORF42-Flag	R	Amplify ORF42-Flag and add EcoRI site	GCGGGAATTCTTAAACGGGCCCCTTGTCGTCG
pJP1-ORF42	R	Amplify ORF42 (untagged) and add EcoRI for restriction enzyme-based cloning	CGAGAATTCTTATTTTGAAAAAAGGGAAACAATGGGGGG
pJP1-ORF42stop	F	Introduce Ser25Stop mutation	CAGGCCTCTTCTCACTCGCGAGTCTTCG
pJP1-ORF42stop	R	Introduce Ser25Stop mutation	CGAAGACTCGCGAGTGAGAAGAGGCCTG
pCDNA4/TO-ORF42 (untagged)	F	Amplify ORF42 (untagged) and clone with Gibson into PmeI-digested pCDNA4/TO vector	CGATCCAGCCTCCGGACTCTAGC
pCDNA4/TO-ORF42 (untagged)	R	Amplify ORF42 (untagged) and clone with Gibson into PmeI-digested pCDNA4/TO vector	GAGGCTGATCAGCGGGTTTAAACTTATTTTGAAAAAAGGGAAACAATGG
ORF42-Flag BAC16	F	Amplify Flag tag, add ORF42 sequence 5’ of tag and remove ORF42 stop codon	CCCCCCATTGTTTCCCTTTTTTCAAAATCAGGGCGGCCGCTCGAGGGA
ORF42-Flag BAC16	R	Amplify Flag tag, add beginning of KanR gene 3’ to the Flag sequence	CTACTTATCGTCGTCATCCTTTAAACGGGCCCCTTGTC
ORF42-Flag BAC16	F	Amplify KanR gene, add end of Flag seq 5’ to KanR	GACGACAAGGGGCCCGTTTAAAGGATGACGACGATAAGTAGGG
ORF42-Flag BAC16	R	Amplify KanR gene, add beginning of Flag seq 3’ to the KanR gene (homologous to sequences at the 5’ end of KanR gene)	TTAAACGGGCCCCTTGTCGTCGTCGTCCTTGTAGTCGATGAACCAATTAACCAATTCTG
ORF42^PTC^ BAC	F	Introduce Ser25Stop mutation	GGAGTGCCAATGAGTACTCATGCCCCGAAGACTCGCGAGTGAGAAGAGGCCTGTCCCGTATAGGATGACGACGATAAGTAGGG
ORF42^PTC^ BAC	R	Introduce Ser25Stop mutation	GGGCACCACAGGGTGGGGGTATACGGGACAGGCCTCTTCTCACTCGCGAGTCTTCGGGGCAAACCAATTAACCAATTCTGATTAG
ORF42^REV^ BAC	F	Revert the Ser25Stop mutation to the wild-type sequence	GGAGTGCCAATGAGTACTCATGCCCCGAAGACTCGCGAGTCAGAAGAGGCCTGTCCCGTATAGGATGACGACGATAAGTAGGG
ORF42^REV^ BAC	R	Revert the Ser25Stop mutation to the wild-type sequence	GGGCACCACAGGGTGGGGGTATACGGGACAGGCCTCTTCTGACTCGCGAGTCTTCGGGGCAAACCAATTAACCAATTCTGATTAG
pCMV-ORF26	F	Amplify ORF26 locus and add sequences homologous to vector for Gibson cloning	GGTTTAGTGAACCGTCAGATCCGCTAGCAGCTAACCCTTCTAGCGTTGG
pCMV-ORF26	R	Amplify ORF26 locus and add sequences homologous to vector for Gibson cloning	TAACGCTTACAATTTACGCCTTAAGTTTTTAATCGTGGTGTAACCAGTG
LANA standard	F	PCR amplification	CCGGTGGAGGTAAAGGTGTTGCGGG
LANA standard	R	PCR amplification	GCAGTCCTGCCTGGGGCACCAATCAG
CCR5 standard	F	PCR amplification	GCACAGGGTGGAACAAGATGG
CCR5 standard	R	PCR amplification	CCCAAGAGTCTCTGTCACCTGCATAG
LANA qPCR [25]	F	Measurement of viral DNA levels	AGGATGGAGATCGCAGACAC
LANA qPCR [25]	R	Measurement of viral DNA levels	CCAGCAAACCCACTTTAACC
CCR5 qPCR [26]	F	Measurement of CCR5 DNA levels	ATGATTCCTGGGAGAGACGC
CCR5 qPCR [26]	R	Measurement of CCR5 DNA levels	AGCCAGGACGGTCACCTT
ORF26 qPCR [27]	F	Measurement of viral mRNA levels	AGCCGAAAGGATTCCACCATT
ORF26 qPCR [27]	R	Measurement of viral mRNA levels	TCCGTGTTGTCTACGTCCAGA
K8.1 qPCR [28]	F	Measurement of viral mRNA levels	AAAGCGTCCAGGCCACCACAGA
K8.1 qPCR [28]	R	Measurement of viral mRNA levels	GGCAGAAAATGGCACACGGTTAC
18S qPCR [29]	F	Measurement of rRNA levels	GTAACCCGTTGAACCCCATT
18s qPCR [29]	R	Measurement of rRNA levels	CCATCCAATCGGTAGTAGCG
ORF57 qPCR [30]	F	Measurement of viral mRNA levels	GGTGTGTCTGACGCCGTAAAG
ORF57 qPCR [30]	R	Measurement of viral mRNA levels	CCTGTCCGTAAACACCTCCG

**Table 2 viruses-11-00711-t002:** Quantification of KSHV protein levels. Average expression and standard deviation of levels of KSHV proteins relative to the WT1 line. *n* ≥ 3 biological replicates—includes data from blots shown in Figure 1C, Figure 2B, Figure 4 and Figure 5.

Viral Gene:		WT1	WT2	ORF42^PTC^	ORF42^PTC^ + ORF42	ORF42^PTC^ + Empty
			Latent			
LANA	latent	1.0	1.0 ± 0.2	0.9 ± 0.2	1.0 ± 0.2	0.9 ± 0.3
Day 4						
ORF6	early	1.0	1.3 ± 0.5	0.8 ± 0.2	1.9 ± 1.3	1.2 ± 0.4
ORF59	early	1.0	1.2 ± 0.3	0.7 ± 0.1	1.1 ± 0.2	0.8 ± 0.2
ORF68	early	1.0	1.3 ± 0.5	1.1 ± 0.9	1.6 ± 1.6	0.9 ± 0.5
ORF26	late	1.0	0.9 ± 0.5	0.6 ± 0.5	1.3 ± 1.0	0.7 ± 0.7
K8.1	late	1.0	1.9 ± 1.0	0.5 ± 0.4	1.2 ± 0.7	0.8 ± 0.5
ORF33	late	1.0	0.7 ± 0.4	0.8 ± 0.8	1.3 ± 0.8	0.8 ± 0.1
ORF52	late	1.0	0.9 ± 0.3	1.1 ± 0.5	2.1 ± 0.6	1.5 ± 0.4
Day 6						
ORF6	early	1.0	0.8 ± 0.6	0.6 ± 0.2	0.8 ± 0.5	0.7 ± 0.5
ORF59	early	1.0	1.0 ± 0.4	0.7 ± 0.4	1.2 ± 0.7	0.8 ± 0.3
ORF68	early	1.0	1.2 ± 0.4	0.5 ± 0.2	0.9 ± 0.4	0.6 ± 0.4
ORF26	late	1.0	1.5 ± 1.0	0.3 ± 0.3	0.9 ±0.8	0.4 ± 0.5
K8.1	late	1.0	1.8 ± 1.7	0.3 ± 0.4	1.3 ± 2.4	0.4 ± 0.6
ORF33	late	1.0	0.9 ± 0.3	0.4 ± 0.1	1.2 ± 0.3	0.7 ± 0.3
ORF52	late	1.0	2.2 ± 2.6	2.0 ± 2.3	3.3 ± 3.7	2.0 ± 2.3

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
