# Peer review of "The Kaposi’s Sarcoma-Associated Herpesvirus Protein ORF42 Is Required for Efficient Virion Production and Expression of Viral Proteins"

_viruses, 2019, doi:10.3390/v11080711_

Round 1
Reviewer 1 Report
Review of revised manuscript Butnaru and Gaglia ORF42 – Viruses
While this revised manuscript has attempted to rectify problems in the original manuscript, it is replete with problems related to data presentation and conclusions with many problems with the written English. There is a lack of understanding of data analysis and determination of significance throughout. There is interesting information in the data presented but it is impossible to accurately assess the data given the way the it is shown in the figures and the way significance is calculated and presented.
Line 2-4 – This revised title doesn’t make sense. What does “expression of some viral proteins” mean??
Line 69-70 –“production of some viral proteins” is vague and meaningless. Consider “This suggests that part of the ORF42 function in KSHV is to promote viral protein expression”. The sentence before indicates what proteins.
Figure 1 – this figure still indicates that the stop mutation is outside of the ORF43 coding sequence and should be fixed to indicate that the mutation affects codons for both ORFs.
Lines 176-178 – as discussed below, this should be shown as viral genomes/ml NOT particles/ml.
Line 273,274 – the absence of shorter ORF42 variants, shown in the western blot , should be indicated here. Consider “This mutation would abolish expression of full-length ORF42. Although other truncated forms starting at later in-frame methionines (for example Met 58) could still be produced, our studies with C-terminal FLAG-tagged constructs, described below, show no evidence of this”. Or something like this sentence.
Line 284 – only one termination codon is studied. Consider “Premature termination of KSHV ORF42 does not….”
Lines 294-296 – the title to panel D “ORF57 mRNA Day” doesn’t make sense. Day??
Line 303 – change “line represent” to “Median titers are indicated”. The statistical indications are not clear – three p levels are indicated “NS, *, and **” yet 4 statistical p values are shown? What is the correspondence?
Line 319 – change to “an early lytic gene, ORF57, …”
Delete lines 325-327 – not clear why this is discussed at this point in the manuscript.
Lines 338-339 – Describe the results of this experiment correctly. “The KSHV ORF42Ptc produced less infectious virions than either WT-infected control, however, a significant difference was only seen in comparison to the WT1 control”. The figure must have bars to show which lanes are being compared for significance. Otherwise it is not clear.
Lines 341,342 – change “ORF42 mutations compromise viral replication” to “ORF42 plays an important role in viral replication”. Multiple ORF42 mutations are not studied here.
Lines 346-347 – since infectious virus, albeit at low levels, is still detected with the ORF42 mutant, ORF42 is not required for completion of the lytic cycle – however it appears to play an important role in enhancing lytic replication– change sentence.
Line 348-351 - delete “natural”, no evidence that variability is natural. Multiple clauses in the sentence are not correct “Because we…” and “in order to …” rewrite sentence. Also, it appears that this sentence is saying that ORF42 attenuates KSHV replication – opposite of what happens. Also, ORF42 function is not altered – which implies a new function. Rewrite sentence. Consider “In order to confirm the inhibitory affect of the ORF42Ptc mutation on viral replication, we examined the ability of full-length ORF42 to restore function in cells infected with the KSHV ORF42Ptc mutation. Plasmids expressing untagged ORF42 or a C-terminal Flag-tagged ORF42 were transfected into the infected cells”.
Line 353-354 – delete
Lines 357-359 – the wording “whereas transduction of the empty vector did not” is questionable since it is not clear whether there are significant differences between the lanes – changing the labeling in panel A would show the significant differences between the lanes.
Lines 409-411 – panel A – the indicated significance in the figure is uninterpretable. Bars must indicate which lanes are being compared with the significance indicated above the bar. Use wording similar to that proposed above for lines338-339 to describe differences with the Wt1 and WT2. Is there a significant difference between the ORF42PTC and ORF42PTC treated with empty vector?? IF so this needs to be stated – could indicate an enhanced induction of replication with the transfection reagents – but this was not equivalent to the restoration seen with the full length ORF42. – this statement needs to be couched with the outcome of the significance analysis, especially if there is no significant difference between the ORF42PTc treated with ORF42 and the ORF42PTC treated with empty vector. In fact the increased expression of ORF57 in the transfected cells panel c, suggests that the transfection partially induces the lytic cycle.
Lines 369-370 – should indicate that there appears to be a slight induction of lytic replication in the transfected cells.
Lines 372-375 – repetitious delete complicated sentence. Unneeded.
Line 376 – change “induction of lytic genes” to “induction of the ORF57 lytic marker”. Delete “early” since all analyses were done days after induction not at an early time point, ie 6 or 12 hrs post induction. DNA replication is not an early event, it is a late event. Change wording. Change “not affected by ORF42” to “not affected by the ORF42PTc truncation mutant”. Delete “also”. Also what??
Lines 380 – change “KSHV WT and ORF42PTC during latency, including in the complemented lines” to “KSHV WT, ORF42Ptc and complemented lines during latency”
Lines 381-382 – this conclusion does not appear to be supported by the data and should be rewritten as discussed below.
Lines 387-389 – delete “result”. Virion production was not analyzed. Change to “reduction of infectious virions”
Lines 393-394 – this sentence does not make sense. Also change “in the levels of cell-free viral particles” to “in the levels of cell-free viral DNA”. That is what is being measured.
Lines 400-401 – the DNA data “suggests” that ORF42 plays a role in the formation and/or release of
Lines 401-403 – delete. Not clear why this sentence is here, especially referring to data from a figure that has not been discussed or presented. Delete.
Lines 420-421 – does total DNA include cell-associated and cell-free? Should specify what this means.
Lines 421-422 – incomplete sentence. Since the analysis is to show what complementation of the mutant does, this should not be graphed relative to WT1. Plot the viral DNA copy numbers themselves.
Lines 423-424 – significance bars need to be indicated on the figure – it appears that the fold induction of the ORF42PTC treated with ORF42 is significantly higher than the ORF42PTC itself. The correct conclusion from this data is that transfection of ORF42PTc with full length ORF42 increases the level of cell-associated DNA (if this is what was analyzed) nearly three fold (estimated from the graph data) over that seen with the ORF42PTc mutant alone. Why is this data graphed as log fold induction? This appears to be a specious way of trying to show that the induction is similar, when in fact it is not. Change this graph to linear and provide a conclusion that is congruent with the data.
424-426 – viral DNA levels does not equal viral particles!!!. DNA PCR does not detect particles or virions. The text for panel f needs to be completely changed. This panel is just showing cell-free DNA levels. Panel f also needs bars showing what lane is significantly different than another lane.
Line 439-440 – change “required for efficient production of intracellular virions” to “required for efficient production of intracellular virions prior to release from the cell”.
Lines 441-443 – delete. This discussion material should be in the discussion section.
Lines 466,467 – change “main expressed isoform of ORF42” to “main isoform of ORF42 expressed in iSLK cells”
Lines 471-473 – since the ORF42FLAG KSHV is not a wild type virus change “in the wild-type context” to “in infected iSLK cells”.
Lines 473-477 – this does not make sense. These sentences essentially say that because the ORF42Flag virus produces less virus than wild type and the ORF42Flag protein does not rescue viral production as well as untagged virus, that the ORF42FLAG virus was chosen to study ORF42 kinetics and localization. Consider changing these sentences to “We tested the ability of the KSHV ORF42-flag virus to infect iSLK cells and produce infectious virions after induction with DOX. We observed that the KSHV ORF42-flag virus was infectious and produced infectious virions (fig 3e). While the level was less than that seen for the wild type, suggesting some downregulation of virion production, we used this system to examine the expression and localization of the FLAG-tagged ORF42.
Lines 509,510 – while ORF42 would not be involved in reactivation due to late expression, it could still be involved with DNA replication which would occur at the same time frame. Delete “and viral DNA replication”.
Lines 531-533 – change “were minimally decreased in the absence of full length ORF42” to “were slightly decreased in the ORF42PTC infected cells after induction, compared to the WT KSHV infected cells”.
Line 534-536 – Change to “In contrast, the levels of the late proteins, glycoprotein K8.1, the capsid protein ORF26 and the late tegument protein ORF33 were strongly reduced in the ORF42PTC infected cells compared to wildtype. A small decrease in the level of the late tegument protein ORF52 was observed.” The ORF33 is decreased as much as the other two proteins, according to the data.
Line536-538 – this statement is not supported by the data. The day 6 results appear to mirror the day 4 results for all proteins. If there are any differences, it would not appear to be significant.
Lines 538-540 – this statement is patently false according to the data. Empty vector restores much if not all of the protein in some cases. This is probably due to an increased induction of replication by the transfecting agents throughout the experiments and should be discussed.
Lines 543-544 – ORF 42 is NOT REQUIRED to produce any of the proteins. They are all produced in the mutant virus situation, albeit at lower levels. REWRITE.
Lines 580 -panel e of figure 5. – the bars for minus ORF42 appear to have all of the duplicate values at relative expression of 1 (little bumps in the bar). This is incorrectly shown. The expression of the minus ORF42 should show the mean of the individual values WITH the standard deviation of those values. The variation in the ORF42 samples needs to include the variation seen in the minus values. Was this done? The title of this figure is not comprehensible. There is no time course so nothing can be said about preceding. ORF42 appears to upregulate the mean ORF26 protein expression without a significant increase in the mean ORF26 mRNA expression. However, is their a correlation between protein expression and RNA expression for each sample, ie does the low RNA point correlate with a low protein point and a high RNA point correlate with a high protein point???
Lines 549-552 – this needs to be changed in regard to the comments above.
Lines 594 -596 - Since protein expression follows RNA expression, it follows that the changes in protein expression at day 4 are probably due to mRNA changes at day 2 or 3. The statement as written has no relationship to a biological effect and should be modified or deleted
Lines 599-600 – “eliminating any transcriptional contribution” has no meaning. ORF42 can be affecting the CMV promoter as well as other promoters.
Lines 602-603 consider this sentence in regard to the points in comments to line 580 above. Also just because ORF42 doesn’t have a detectable effect on ORF26 mRNA expression by the CMV promoter at the time point chosen doesn’t mean that it couldn’t have an effect on the ORF26 promoter.
Lines 604, 606 – this sentence doesn’t make sense
Lines 638 – Figure 6 panel b – the replicate data for the ORF42 positive cells cannot be represented this way. Show the fluorescence data for each sample with means/medians and show standard deviations.
Reviewer 2 Report
The authors have adequately address all comments from the previous review.
Author Response
We thank the reviewer for their comments on the previous version of the manuscript.
Reviewer 3 Report
I am content with the way my comments were addressed and with the revision of figures and text.
Please note for Figure 1 (d) that the title in the figure says "ORF57 mRNA Day" - the number for the day is missing.
Author Response
We thank the reviewer for the comments on the previous version of the manuscript.
We have now corrected the typo the reviewer pointed out - the panel now says "ORF57 mRNA day 2"
Reviewer 4 Report
1. The answers to questions 1 and 2 taken well.
2. As presented by the authors, Notch signaling has a crucial role in KSHV reactivation. It is imperative to test the effect of inhibiting JAG1/Notch1 signaling on the expression of ORF42 and associated signaling. This will complete the study and at the same time be a starting point for numerous other studies on these lines.
3. Accordingly, a short note on JAG1/NOTCH1 signaling in the introduction or discussion must be included.
Author Response
1. The answers to questions 1 and 2 taken well.
Thank you.
2. As presented by the authors, Notch signaling has a crucial role in KSHV reactivation. It is imperative to test the effect of inhibiting JAG1/Notch1 signaling on the expression of ORF42 and associated signaling. This will complete the study and at the same time be a starting point for numerous other studies on these lines.
We are still not sure how testing this pathway would provide additional information in this particular study. All the literature on JAG1/Notch indicates a role in controlling reactivativation and early lytic gene expression. However, viral reactivation is not impaired in the ORF42PTC-infected cells and ORF42 is expressed with late kinetics, after the lytic cycle is presumably well under way. Therefore, one would not expect any specific interactions between ORF42 and the Notch pathway, even at the level of ORF42 expression. Moreover, we are not focusing on the regulation of ORF42 expression in this study, only its activity, which appears to be unrelated to reactivation. Lastly, given that we were given only a few days to modify the study for resubmission, we are unable to test the pathway at this time.
3. Accordingly, a short note on JAG1/NOTCH1 signaling in the introduction or discussion must be included.
Since we are not focusing on the process of reactivation in this study, we do not extensively discuss anywhere the dynamics and control of KSHV reactivation. Therefore, we are not really sure how we would integrate this information in either our introduction or discussion. We would be happy for the reviewer to suggest more specifically where this information would fit.
This manuscript is a resubmission of an earlier submission. The following is a list of the peer review reports and author responses from that submission.
Round 1
Reviewer 1 Report
MAJOR QUESTIONS:
Did the authors test their virus on physiologically relevant cells like human B cells, or endothelial cells? This is very important information.
Fig. 3: Why did the authors use 293 cells to monitor infection of the virus? 293 cells are not appropriate cell type to study KSHV infection of cells. They MUST use physiologically relevant cells like HMVEC-d cells or B cells.
What is the effect of JAG1/Notch1 inhibitors on the expression of ORF42 and associated signaling?
Reviewer 2 Report
This manuscript details an extensive analysis of the function of KSHV ORF42 using both recombinant viruses in which an amino acid codon has been changed to a stop codon to block ORF42 expression and viruses expressing recombinant ORF42 to test for regain of function. The authors have not accounted for the particular genomic structure of KSHV in the region of ORF42, which could seriously affect most of the conclusions drawn in the manuscript, as detailed below. While the authors refer to their mutant as a deltaORF42 KSHV, implying that ORF42 was deleted from the recombinant virus, in fact, they have only altered one codon to eliminate the predicted AUG initiating codon. No experiments were done to prove that ORF42 or truncated versions of ORF42 were not produced in the deltaORF42 virus. This is a serious flaw given that the predicted expression of the ORF43/42 bicistronic locus would produce a natural truncated version of ORF42 from either of two possible AUG initiators that remain intact in the deltaORF42 KSHV. The authors must either confirm the presence or absence of ORF42 protein products in their experiments or completely revise their text to eliminate all reference to “loss of ORF42” in their results and discussions. Instead indicating simply that the result was obtained with the SER25Stop mutant. In the discussion, the caveats for these results would be presented with a complete discussion of the predicted and potential gene expression occurring in this locus. There are many interesting results from this extensive investigation, however, their impact is not clear due to the unknown nature of the gene expression in this locus.
Line 64 – sentence is a bit awkward, possibly change “cytoplasm that, like” to “cytoplasm, and like”
Line 104 – it is important to state here that the ser to stop mutation (presumably TCA to TGA) also changes the Valine codon in the overlapping ORF43 from GTC to GTG, which does not alter the encoded protein.
Line 107 – It is important to state, if correct, that insertion of the FLAG sequence downstream of ORF42 repositioned the AATAAA polyadenylation signal for the ORF42 transcript, which is encoded at the end of ORF41, to a position downstream of the inserted FLAG sequence, thereby insuring that processing of the ORF42 transcript was not affected.
Line 117 – important to note here that the alteration in the position of the AATAAA polyadenylation signal after the FLAG sequence was confirmed by sequencing – a supplementary figure showing gene expression and position of regulatory regions within the ORF42/43 locus would be helpful.
Line 130 – should add “Since we analyzed only one clone for both the delta ORF42 and ORF42-FLAG recombinant viruses, we were unable to determine variability in the phenotypes of the recombinant viruses”.
Line 150 – it is not clear why viral titer was calculated this way, since the level of uninfected cells remaining at day 6 is being used in the calculations and these cells could have been replicating during the 6 day period. How does replication of the uninfected cells affect the titer calculations?
Figure 1 – The figure showing the position of the S25STOP mutation is incorrect. ORF43 overlaps the beginning of ORF42 and the S25STOP mutation affects both ORFs, as indicated above. This should be redrawn to indicate this. It is not clear that the terminology of “deltaORF42” is correct.
The proposed model of the transcription and translation of wild type ORF42 should be introduced in the introduction. ORF42 and ORF43 form a bicistronic locus, in which the only known polyadenylation site is positioned downstream of ORF42. Therefore transcripts initiating at the presumed promoter for ORF42 (located at about bp 63451 (TATTTATAT) in the NC_009333 reference sequence) would produce a monocistronic transcript encoding ORF42 and ending at the AATAAA (bp 62539-62534; NC_009333) which contains part of the terminal Lysine codon (AA) and the TAA stop codon for ORF42. The ORF42 AATAAA sequence is complementary to the terminal sequence of ORF41 encoding the 3 C-terminal amino acids. The transcripts initiating at the promoter for ORF43 would encode ORF43 and ORF42 in a bicistronic transcript. Since ORF43 and ORF42 overlap, ribosomes translating ORF43 on the bicistronic transcript would either disassociate from the transcript after the ORF43 coding sequence or would continue scanning the transcript for the next AUG sequence to initiate further protein translation. Surprisingly, the next AUG sequences after termination of ORF43 are two possible AUG initiators 38 (bp 63197; NC_009333) and 47 bp (bp 63188; NC_009333) downstream of the ORF43 stop codon. Since the ORF25STOP occurs within the overlapping region of ORF42 and ORF43 upstream of these two potential AUG initiators, truncated versions of ORF42 could still be translated from the bicistronic ORF43/42 transcript in completely biological fashion. This suggests a potentially troubling scenario for the subsequent experiments using the deltaORF42 KSHV. In the case of the monocistronic transcript from the ORF42 promoter, the deltaORF42 KSHV sequence would encode an upstream uORF ie, the first 24 aa of the ORF42 N-terminus which could regulate the translation of a truncated ORF42 sequence initiating from one or both of the AUGs at bp 63188 and bp 63197 and encoding the majority of ORF42. In the case of the bicistronic transcript from the ORF43 promoter, the deltaORF42 KSHV sequence would specify the natural bicistronic transcript encoding the natural ORF43 sequence upstream and the natural truncated ORF42 sequence downstream. The whole manuscript should be reconsidered with these possibilities in mind. If there are data to suggest that this altered translation of truncated ORF42 is not occurring, then this should all be discussed in the Discussion. It is even possible that full length and truncated ORF42 have different natural biological functions. The positioning of potential alternate AUG initiators in ORF42 just downstream of the termination site of ORF43 is conserved in RFHVMn, the macaque homolog of KSHV, and EBV suggesting that this scenario of gene translation is important.
Line 235 – period missing after induction.
Line 237 – what level of induction of ORF57 mRNA was obtained in the WT1 line after 2 days? Does this timing truly reflect robust induction? What is the experimental variation for this data and was the variation of the WT1 ORF57 expression part of the significance calculations. This should be stated in the materials and methods. Since the variation in the data for both WT1 and WT2 represents the variation in the WT experiments, shouldn’t the comparison to the deltaORF42 be done to the combination of both WT1 and WT2. This goes for all of the subsequent analyses with both WT1 and WT2 – It is not clear what the reasoning is for choosing one or the other of the WT samples to normalize to or to do the statistics.
Line 242 – this is inaccurate – as sequence homology with related viral proteins indicates a function – please rewrite.
Line 244 – what was the latent copy number of WT1 which was used for normalization?
Line 250 – annotations is not the correct word here – conservation of sequence predicts function. This and the previous sentence are the basis for this project and should be accurately written.
Line 252 – given the discussion above, this lacking expression part of this statement should be reconsidered. Just indicate that serine 25 was replaced with a stop codon to block ORF42 expression. Whether the expression was actually blocked is another issue that would be discussed in the discussion.
Line 254 – Figure 1A does not reflect this sentence, since the mutation does not appear to be within the ORF43 coding sequence.
Line 259 – please accurately describe the iSLK.RTA cell line
Line 263 – the possibility that similar variation could occur in the induction of the mutant cell lines needs to be addressed here.
Line 273 – what is the evidence that the 2 day time frame after DOC activation is a measure of robust reactivation? What is the level of ORF57 expression prior to DOC activation and how much did this increase after activation?
Line 274 - this statement is incorrect – the data show that the deltaORF42 infected cells with ORF42 overexpression had nearly double the amount of ORF57mRNA compared to wild type and deltaORF42, but this difference was not significant due to the large variation in RNA levels detected in replicate experiments (this appears to be 3 from the legend). The most important part of this experiment was to determine the fold induction of ORF57 mRNA after induction. This is the measure of activation, not the absolute level of ORF57 mRNA compared to wild type. No conclusion can be made from this experiment.
Line 277 – this statement is inaccurate – In Figure 1d, the first important comparison is the level of viral DNA in the different latent cell lines – this cannot be clearly observed in the figure. The second most important comparison is the increase in viral DNA in each cell line after activation – this is not analyzed in the figure. For example, the statistical analysis is not between the level of DNA after activation of the WT1 and deltaORF42, which was not significant. The analysis should be between the fold induction seen in the WT cells and the fold induction seen in the delta ORF42 or the deltaORF42+ORF42. There would appear to be a significant increase in the induction of viral DNA in the deltaORF42 cells compared to wild type. Furthermore, the overexpression of ORF42 significantly increases the level of viral DNA in the deltaORF42 cells, suggesting that the ORF42 provided in trans is biologically different than the ORF42 provided in cis in the wild type. See discussion in point 170 above. The large variation in DNA copy number for the WT cells is very troubling and suggests uncontrolled factors affecting reactivation. This large variation makes the comparison with the delta ORF42 unclear. The increase in viral DNA after induction would appear to be a measure of the induction of viral replication.
Line 278 – this is not an accurate statement – the increase is not slight!
Line 281 – this needs to be rewritten given the points raised above. It is not clear what functionality of the BAC refers to - rewrite. It is also incorrect to say that ORF42 is dispensable for KSHV reactivation from latency when the only measure is ORF57 expression and the results do not even show the increase in ORF57 expression from latency.
Line 283 – viral replication can indicate two different things – either - replication of the viral DNA – which was analyzed in Figure 1D or production of infectious virions – the results from 1D show that the genetic alteration in delta ORF42 does not inhibit viral DNA replication. Clearly, addition of wild type ORF42 in the presence of deltaORF42 enhances the viral DNA replication after induction over that seen in all of the other different experiments. A western blot analysis of ORF42 expression targeting a C-terminal domain is needed to prove that ORF42 expression is blocked.
Line 292 – states that deltaORF42+ORF42Flag rescued virus production, yet in Figure 2a it is indicated that this is non significant. It would seem that to determine whether virus production was rescued that the level of virus in the supernatant of the different samples should be compared to wt and not to deltaORF42. Furthermore, it is not clear that the mean should be used in this figure as opposed to the median titer. It appears that ORF42 in trans does not completely rescue virion production
Line 318 – this statement is not correct given the discussion of Figure 1 above.
Line 333 – this figure is not clear – what are the duplicated regions and how do they affect the polyadenylation site – put a description in supplemental data.
Line 338 – are the days post induction? Not clear. This experiment is not adequately described in the materials and methods. When was PAA added? How much?
Line 345 – lysates were analyzed by SDS gel electrophoresis and proteins were detected by western blot. Lysates were not stained.
Line 360 – this is not clear
Figure 4a – it is not clear that the ORF6 expression in WT2 is 0.8 the level of WT1
Line 402 – this does not accurately reflect all the data - see ORF52 day 4.
Line 404 – this may be due to the presence of truncated ORF42, as discussed above.
Line 410 – since there is no data to indicate that a natural truncated version of ORF42 is not expressed – it is incorrect to say “loss of ORF42” – needs to be rewritten for accuracy. Either use current data and say “S25 stop mutation in ORF42 impairs the accumulation ..” Or perform additional experiments showing the absence of the natural truncated ORF42 protein.
Line 415 – since at least 3 biological replicates were done for these experiments, the relative levels shown should be the average from these experiments with standard deviation? This would provide strong confirmation that the blots shown were representative.
Line 422 – change “protein and mRNA was collected” to “were collected”.
Line 424 – punctuation not clear. Why comma after 8? What is the variation in the 8 data points for WT1 – this should be shown and is this variation used in calculation of the significance? What is the variation in the latency proteins in the different cell lines before activation? Does the deltaORF42 KSHV produce lower levels of all proteins regardless of activation? Can the difference in protein levels after activation be attributed to an inherent difference in protein levels in the different cells before activation and has no relationship to activation? This issue should be discussed. If the deltaORF42+empty is the control for deltaORF42+ORF42, shouldn’t the statistical analysis show the significance of the difference between these two and not with deltaORF42 itself?
Line 429 – how was the level of 18S rRNA calculated for the normalization?
Line 435 – it is not clear that this statement is correct given the lack of data on the variability of the WT. The level of difference should be indicated because this was quite small and barely significant. In fact, no decrease was evident between WT and the deltaORF42+empty.
Line 437 – for deltaORF42?? Not stated
Line 438 – This statement is inaccurate. The data indicate no significant change in the protein levels of ORF26 on Day 4 and no significant change in the mRNA levels of ORF26 on Day 6.
Line 440 – given that the variation in the untreated sample is not shown, the significance of these data is not clear.
Line 443 – this statement is not clear given that the ORF26 expression is driven by a CMV promoter which does not exist in the KSHV genome. What the relevance of this experiment is to the intact KSHV is not clear as well.
Line 452 – see discussion above about presence or absence of ORF42, since this was never measured.
Line 459 – it is well known that KSHV activation reduces cellular protein expression, therefore it is not clear that the increased fluorescence is due to cellular proteins.
Reviewer 3 Report
The manuscript by Butnaru and Gaglia, The Kaposi's sarcoma-associated herpesvirus protein ORF42 is required for efficient viral replication and viral protein accumulation is a preliminary study designed to address the role of the previous uncharacterized lytic protein ORF42. Based on its homology to other herpesvirus proteins, ORF42 has been classified as a tegument protein. The experiments in the manuscript demonstrate that ORF42 is not required to reactivate KSHV from latency but is required for efficient accumulation of late viral proteins and is suggested to potentiate post-transcriptional stages of viral gene expression. In other herpesviruses lacking the ORF42 homologues, the decrease in virus production is related to defect in virion formation and viral egress, and there is data that suggests additional functions of ORF42.
Minor points:
The figures with statistically significant p-values (figure 2). It seems a little awkward to have the p-values relative to the deltaORF42 virus, normally this is reported as relative to the wildtype viruses (as is reported in Figure 5).
There is no description in the material and methods for the isolation of the nuclear and cytoplasmic fractions in figure 3G.
Figure 6. It's unclear how ORF42 is affecting the rate of cellular protein synthesis and is complicated by using cells that are infected and undergoing lytic reactivation. To exclude the possibility of other viral proteins contributing to the increase in the rate of cellular protein synthesis (or perhaps decrease the rate of protein decay) they should to the metabolic labeling in the absence of infection.
Reviewer 4 Report
1. General summary and opinion about the principle significance of the study, its questions and findings
Butnaru and Gaglia study the role of the KSHV protein ORF42 for the viral life cycle. Not much is known for this herpesviral core protein, and sequence homology with its homologs in other herpesviruses (HSV-1/-2 UL7, HCMV UL103) is low. ORF42 has previously been classified as a late gene, and based on studies on HSV-1 UL7, ORF42 may be a tegument protein. For HCMV UL103 it has been shown that it plays a role for biogenesis of the cytoplasmic virion assembly complex, the site of final virion assembly.
The authors generate a KSHV mutant lacking ORF42, and characterise this mutant in vitro.
Overall, the question raised in this study is justifiable, and the findings are interesting, but not groundbreaking. However, I have major concerns with the study as outlined below.
2. Specific major concerns essential to be addressed to support the conclusions
1. I am confused with Figure 1 and the corresponding result section. This needs to be rewritten for clarity and even restructured. The scheme of Figure 1a is fine. But what does the agarose gel in Figure 1b tell me, meaning: What is the result of it? Why do the authors show it at all? Actually, common standard these days is to completely sequence new mutants. This costs about 250 US dollars and shows if other mutations were introduced by mistake during the recombination process. Or a revertant should have been included in the study. Without a revertant or without complete sequencing I cannot recommend to accept this study.
Figure 1c shows ORF57 mRNA levels (ORF57 is an early gene) in infected iSLK.RTA cells. Levels are presented as relative to WT1, and shown are only the ORF57 mRNA levels in cells that were reated with Dox to induce RTA expression and thereby the lytic viral life cycle. I would also like to see ORF57 mRNA levels in cells that were not treated with ORF57 to judge if the induction worked properly.
Figure 1d shows latent and lytic viral copy number of LANA, which is a latent gene. Here, an additional truely lytic gene should be included, such as ORF57. Why did you measure CCR5 as the housekeeping gene? Please specify. I am actually not sure what this figure tells me. It also does not fit to the title "Generation of deltaORF42 virus", it is already a characterisation of the mutant. See also minor comment 6. The legend for this figure is also not written very clearly.
Also, complementation assays are performed here (Figure 1c and 1d) - this should be clearly stated since they are important. Actually, what should be shown in Figure 1 show be growth curves comparing WT and mutant virus - this is usually the first experiment to be done after a new mutant has been generated.
Result section line 273: the authors conclude that ORF42 is not important for reactivation. To my knowledge, the KSHV field asks for a proper reactivation assay; I am not sure that the assays presented in Figure 1c and 1d meet the current standard.
In summary, I am missing a "red line" for Figure 1 and in their rebuttal the authors should justify why they chose those experiments.
2. Line 296-298: hypothetical - it may also just be expressed at lower levels. This should be moved and discussed in the discussion.
3. Line 316-317: Egress per se may not be affected at all.
4. Figure 3b: What is the result of this gel? As noted earlier, the mutant needs to be sequenced completely. Figure 3c: Lane 1 is not labeled.
5. Figure 3E needs a quantification as done for Figure 2a.
6. In general, the study lacks analysis of endogenous ORF42. Since an antibody is not available, qPCR could be used to study expression kinetics of ORF42, this would also nicely add to the characterisation of the mutant virus (albeit mRNA may be detected in the mutant depending on the primers chosen).
7. Conclusion for Figure 5: What exactly is the conclusion from this figure? It may help to work with the word significant - protein level differences were highly significant at day 6, and RNA levels were also significantly affected. I am not sure one can directly compare those two assays and say: effect on protein is higher than on RNA levels. Conclusion for Figure 5e: I would be careful here, the assay is quite artificial (overexpression, CMV promoter).
8. Figure 6: I am not sure whether I understand this set up. Cells either express ORF42 or empty vector (I assume more than 90% since they are under selection, but ideally this would be shown - if possible - by Immunofluorescence by the authors). These cells are infected with KSHV deltaORF42. Then the lytic cycle is induced. 5 days later, cells are labeled to measure protein synthesis. The authors note in line 454-456, that KSHV may not reactivate in all infected cells, and that this may explain the small difference shown in the figure. This makes no sense to me, or I may simply not understand the experiment: all cells should express ORF42 (left dataset), or none (right dataset). Upon RTA expression, KSHV should reactivate, but the virus does not express ORF42. So if ORF42 has an effect on cellular protein synthesis, I should see this clearly, since all cells express it in the left dataset (I assume so at least).
The correct way of doing this assay - in my opinion - would be to infect iSLK.RTA cells with either WT or deltaORF42 KSHV, then label the cells before ORF42 is expressed (and this time point should be determined by qPCR beforehand), and then do FACS to measure AHA incorporation. Uninduced cells should serve as control since they should not express ORF42.
3. Minor concerns that should be addressed
1. The title is misleading:
a) ORF42 is required for viral replication -> To make this conclusion, I would expect to see a growth curve. The ORF42 mutant should be impaired in its growth compared to WT KSHV.
b) ORF42 is required for viral protein accumulation -> accumulation may mean that proteins are not transported to the right location and therefore accumulate in specific compartments. I would rather write: "....required for expression of selected viral proteins". See also legend for Figure 4.
2. References are missing for the statement in lines 47-49: "based on findings in other herpesviruses, it is classified as a component of the tegument"
3. References are missing for the statement in lines 62-63: "In gammaherpesviruses, ORF42 and its homologs....." Please name the homologs, gammaherpesviruses, and give references.
4. The effect of ORF42 on expression of late proteins has only been analysed for a few, and not all of them were affected. So please do not generalise results as e.g. in line 67-68.
5. The mutant virus is called deltaORF42. It is rather a stop mutant though, and since no antibody exists that detects ORF42 protein in the context of infection it cannot be concluded that no protein is made - a truncated version may be translated from start codons after the stop codon insertion. Therefore, I would recommend to call the mutant virus ORF42stop. Also, please give more details why the stop cassette was inserted at the site chosen, and if any in frame methionines occur afterwards.
6. Subheadings: 3.1 also includes data about reactivation - not only about generation of the mutant virus. Please insert approbiate subheadings.
7. line 235, legend of figure 1: "and" is missing
8. line 367: "as evidenced by"